# Learning domain-specific causal discovery from time series

**Xinyue Wang**                      *wsinyue@seas.upenn.edu*
*Department of Bioengineering*
*University of Pennsylvania*

**Konrad Kording**                      *koerding@gmail.com*
*Department of Bioengineering*
*University of Pennsylvania*

**Reviewed on OpenReview:** *https: // openreview. net/ forum? id= JFaZ94tT8M&noteId= 3yzZwnsMng*

## Abstract

Causal discovery (CD) from time-varying data is important in neuroscience, medicine, and machine learning. Techniques for CD encompass randomized experiments, which are generally unbiased but expensive, and algorithms such as Granger causality, conditional-independence-based, structural-equation-based, and score-based methods that are only accurate under strong assumptions made by human designers. However, as demonstrated in other areas of machine learning, human expertise is often not entirely accurate and tends to be outperformed in domains with abundant data. In this study, we examine whether we can enhance domain-specific causal discovery for time series using a data-driven approach. Our findings indicate that this procedure significantly outperforms human-designed, domain-agnostic causal discovery methods, such as Mutual Information, VAR-LiNGAM, and Granger Causality on the MOS 6502 microprocessor, the NetSim fMRI dataset, and the Dream3 gene dataset. We argue that, when feasible, the causality field should consider a supervised approach in which domain-specific CD procedures are learned from extensive datasets with known causal relationships, rather than being designed by human specialists. Our findings promise a new approach toward improving CD in neural and medical data and for the broader machine learning community.

## 1 Introduction

Decision-relevant insights are usually about causation. In public health, practitioners want to decide policy interventions based on causal relations (Glass et al., 2013). In sociological research, scientists want to use causal language to answer what effect a specific behavior can have (Sobel, 1995). In biomedical science, researchers ask what causal mechanism a new medicine has (Imbens & Rubin, 2015). In neuroscience, we may ask about the causal role of a neuron's activation or a lesion in a brain region (Marinescu et al., 2018). Causal insights are at the heart of science, engineering, and medicine.

To uncover the causal relationships inside a complex system, researchers often start by estimating the causal influence from one element to another. Randomized controlled trials (RCTs) are regarded as the gold standard in clinical and societal problems (Zheng et al., 2020; Sherman et al., 2005). However, RCTs can be expensive, ethically challenging, and often infeasible. Therefore, revealing causal relationships from observational data, known as *causal discovery*, is an active field of causality research. Relying on different assumptions, causal discovery methods can be broadly categorized into conditional-independence-based, structural-equation-based, and score-based methods (Spirtes et al., 2000b;a; Shimizu et al., 2006; Hoyer et al., 2008; Zhang & Hyvärinen, 2010; Chickering, 2002; Huang et al., 2018; Zheng et al., 2018; Zhu & Chen, 2019; Goudet et al., 2017; Ke et al., 2022). While these methods are developed for general causal discovery, time-series data provides additional means for causal discovery. Therefore, a separate branch of methods has been developed specifically for time-series data.

Time-series causal discovery methods generally uncover causal relationships by leveraging human domain knowledge. Granger Causality, a well-known method, relies on statistical tests and linear regression under the assumption that causes precede effects (Granger, 1969; Lütkepohl, 2005). Extensions of Granger Causality consider multiple variables, non-linearity, and history-dependent noise, using vector autoregressive models (VAR) and lasso penalties (Lozano et al., 2009; Shojaie & Michailidis, 2010). Model-free variations, such as transfer entropy and direct information, capture non-linear causality (Vicente et al., 2011; Amblard & Michel, 2011). Neural network-based extensions model non-linear temporal dependencies and history-dependent noise (Tank et al., 2021; Nauta et al., 2019; Khanna & Tan, 2019; Löwe et al., 2022; Gong et al., 2022; Bi et al., 2023). Conditional-independence-based methods identify partially directed acyclic graphs (DAGs) by iteratively testing conditional independence, assuming causal sufficiency and faithfulness. Fast Causal Inference (FCI) has been extended to time-series data Spirtes et al. (2000c), and the PC algorithm has been combined with momentary conditional independence (MCI) tests (Runge et al., 2019; Runge, 2020; Gerhardus & Runge, 2020). Score-based methods, such as DYNOTEARS, tackle time-series causal discovery as an optimization problem on dynamic Bayesian networks (Zheng et al., 2018; Pamfil et al., 2020). Structural Equation Models (SEM)-based methods, such as Linear Non-Gaussian Acyclic Model (LiNGAM), have also been adapted for time-series data by incorporating properties like latent variables, non-stationarity, and non-linear effects (Shimizu et al., 2006; Hyvärinen et al., 2010; Rothenhäusler et al., 2015; Huang et al., 2019; Peters et al., 2013). The field of causal discovery from observational time-series data has gained popularity in machine learning and statistics (Moraffah et al., 2021; Pearl, 2010; Schölkopf et al., 2021; Glymour et al., 2019), implementing heuristics implied by domain experts.

Despite having rigorous theoretical justification, existing causal discovery methods critically rely on strong human assumptions about causality and aim to work across domains (see Figure 1.B). If these assumptions are correct, then we can use them to construct good inference algorithms. However, it is unclear whether they can still work well when the strong assumptions behind them are not guaranteed in the real world. Human intuition may be limited, and as history has shown, humans often lack complete intuition (Sutton, 2019). Consequently, we need to explore whether learning can surpass human ingenuity in the causality domain, as it has in other fields.

Contrary to causal discovery methods, ML has achieved rapid progress in numerous domains by building estimators in a data-driven way (see Figure 1.A). It primarily relies on patterns in the data rather than explicit human assumptions. Leveraging abundant data and computation power, ML techniques have accomplished remarkable feats in fields like text generation, image generation, face recognition, and games (Ouyang et al., 2022; Rombach et al., 2022; Lu & Tang, 2015; Berner et al., 2019; Silver et al., 2016; Vinyals et al., 2019). For causal discovery, a domain-specific causal estimator should be generated in a data-driven manner without hand-engineering. With ample causal ground truth data, modern ML techniques, including neural networks, could potentially enable efficient learning of time-series causal discovery.

Here we consider learning a domain-specific estimator for time-series causal discovery that is trained and applied to time-varying data of a specific domain under the i.i.d. assumption across problem instantiations (see Figure 1.C). More specifically, we assume to have a training set of causal discovery problems where we know causal ground truth. This is then used to learn a causal discovery strategy, through supervised learning where each sample is a dataset. We use supervised learning on large datasets of known causal relations to discover an algorithm that will correctly identify causality from unsupervised observations. To implement this algorithm, we use a standard transformer and compare it to other methods. We thus construct a system that can learn to infer causal relationships inside a specific domain. Then we use that learned causal inference strategy and see how well it predicts ground truth on the test set of causal discovery data sets.

We apply our learning procedure to several standard datasets and a newly generated one. Our main example application is a newly generated dataset of causal relations from the microprocessor MOS 6502, a deterministic complex system with many causal components. We identify causality using perturbations of all unique transistors. We thus know which transistors affect which other transistors.

Our contributions in this work are as follows:

- We use learning to generate an estimator for domain-specific causal discovery on time series data.

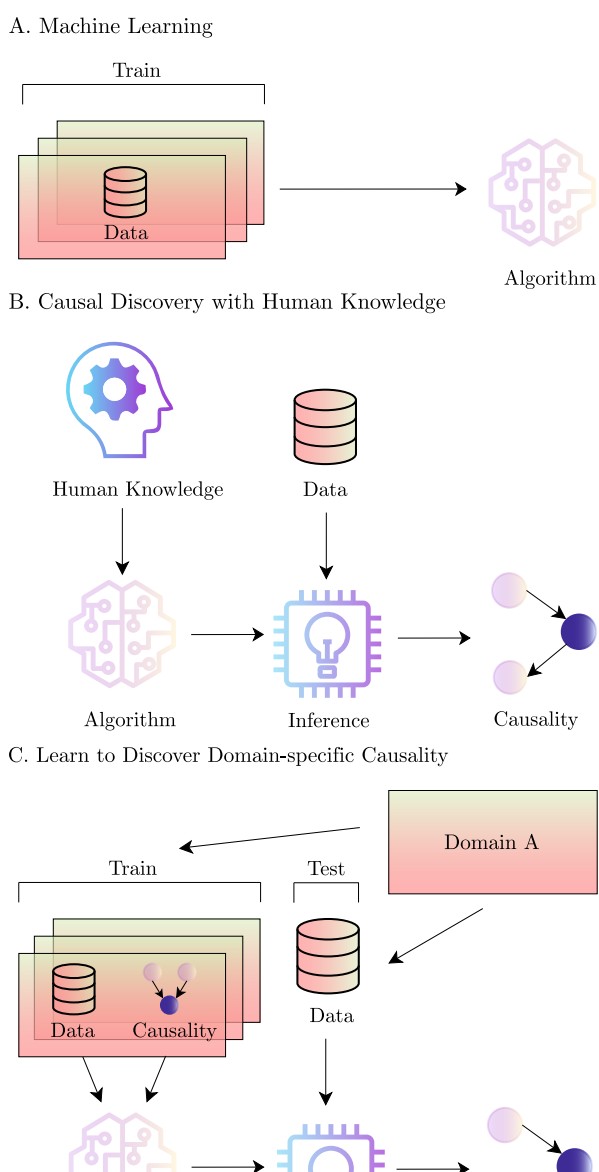

Figure 1: **A.** In machine learning, estimators are generated from data and then used in inference on unseen data. **B.** In traditional causal discovery procedures, we define causality in the causal discovery algorithms by human knowledge rather than driven by data. These estimators infer causality in unseen data by seeing how well the data fit our assumptions. **C.** Similar to machine learning, domain-specific causality estimators are generated from lots of observational data and known causality from the same domain, through maximum likelihood estimation. We only expect them to learn the causality and to be applied in the same specific domain, with fewer and weaker human assumptions, in a data-driven way.

- We test it on various domains, including the microprocessor 6502, a simulation of fMRI signals, and gene networks. We show that in these three domains, it works better than algorithms based on human intuition.

- We examine the generalization of our learned estimator under different levels of small-scale noise and different behaviors of the system.

- We conduct an explanation study by Grad-CAM on the last attention layer of our Transformer-based architecture and find that our procedure has helped the estimator learn the causality of a specific domain.

## 2   Methods

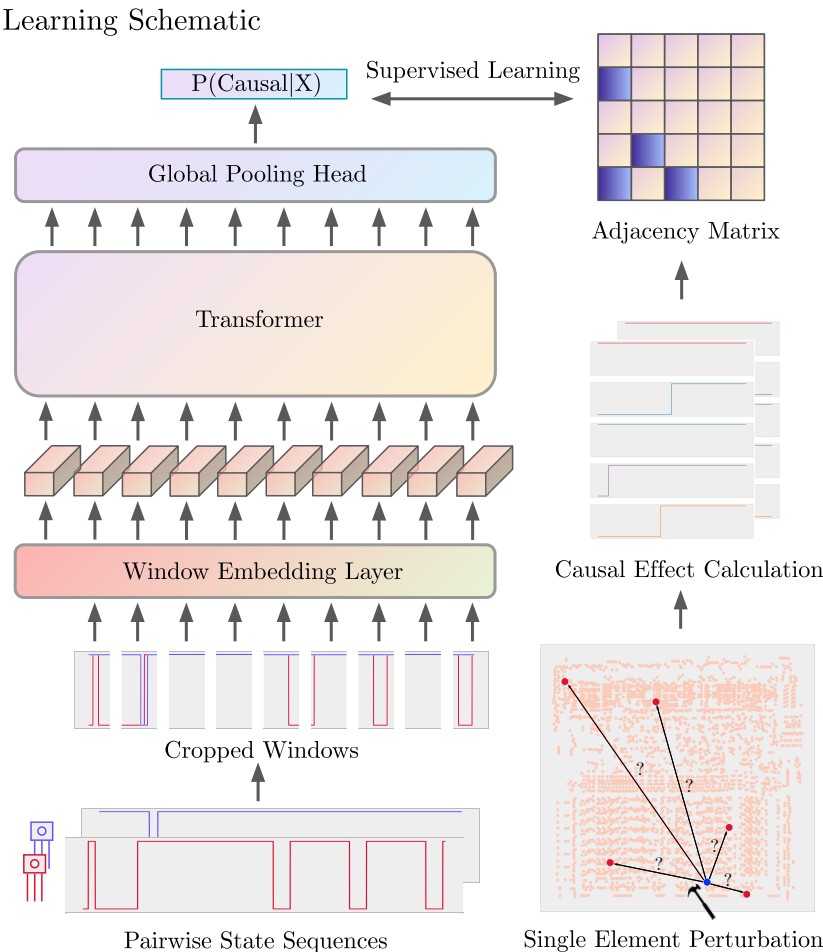

Figure 2: Schematic for learning domain-specific causal discovery from time-series. On the left is the causal discovery model structure, we use transformer-based architecture as the sequence encoder. On the right is the perturbation workflow used to get causal ground truth inside the microprocessor.

To be able to test algorithms of causal discovery, we need to have a database of known causal relations. We, therefore, use a deterministic system that contains many causal relations, the MOS 6502 Microprocessor. We can readily measure causal effects by perturbing transistors and seeing how this affects the voltages of other transistors in a short time (one half-clock). We can then ask if an algorithm that is trained on a part of the microprocessor can be used to infer causal influences inside the other part of the microprocessor.

In order to know about causality, we need to start with the perturbation experiment of the MOS 6502. We take a C++ optimized MOS 6502 simulator (Jonas & Kording, 2017) with three game recordings (Donkey Kong, Pitfall, Space Invaders) as the target system. We first acquire the causal relationship among transistors by single element perturbation analysis and define it as the ground truth of the causality inside the MOS 6502 system, which is then utilized as a supervised signal and a validation standard in our causal discovery procedure (we could have used the netlist but that one has no directions distinguishing cause and effect). This way, we know the ground truth causal influence of transistors upon one another, which we can use for learning a strategy and for testing.

## 2.1 Single element perturbation analysis

Identifying causal relationships in the MOS 6502 microprocessor solely from observations is challenging. The MOS 6502 is a large, complex system comprising 3510 transistors and 1904 connection elements, which facilitate the formation of multi-input and multi-output (MIMO) connections among transistors. Although the existing netlist provides information on connection elements, it lacks directional information and details about the paths through which MIMO connections flow. Consequently, it is difficult to draw conclusions about the cause-effect relationships one transistor has with others. To uncover causal relationships among transistors, we conduct single element perturbation experiments, an approach commonly used to infer causality in brain networks, cellular events, and genetics (Paus, 2005; Meinshausen et al., 2016; Welf & Danuser, 2014). The main idea is that when a causal element is perturbed, its downstream targets will be affected, and we can determine the causal effect by comparing it with unperturbed conditions. By calculating the difference between perturbed and unperturbed states, we can ascertain if a causal influence flows from cause to effect.

**Temporal precision** is critical for capturing causality in time-varying data, particularly in the presence of unobserved confounders. Low temporal precision may obscure the temporal differences between cause and effect at the same time point, making them challenging to distinguish. Hence, we increase the sampling rate to allow sampling within every half-clock, providing a higher temporal resolution of current flows among transistors. Specifically, we define the update steps within a half-clock as $k$ and use 30 as $k$ in our experiment. When the within half-clock update steps are fewer than $k$, we pad it with the last state to $k$. For the rare cases where the within half-clock update steps exceed $k$, we truncate them to $k$. Then, we concatenate all the within half-clock sequences to construct the full recording of the microprocessor. For instance, a 128 half-clock runtime recording would result in a total sequence length of 3840 steps for a transistor, where the update step is the smallest unit in the transistor state recording.

**High repeatability** is a common but tricky phenomenon that occurs in ideal systems. Due to MIMO connections, it is typical for some transistors to have nearly identical state sequences in a short time. To reduce redundancy in the perturbation experiment and causal discovery procedure, we remove duplicates every time we collect the full recordings of all transistors, the quantity of which varies across games and time. We thus only analyze transistors exhibiting unique behavior.

**Sparsity** is a prevalent property of causality in real-world systems. Compared to non-causal relationships within a system, causal relationships are sparse and become increasingly so in larger systems. For example, during the period from half-clock 0 to 128 in the microprocessor system, there are only around 300 causal pairs out of a total of $36k$ transistor pairs. Similar to repeatability, this quantity fluctuates across games and time. We, therefore, conduct multiple perturbations at different time periods to collect a large number of positive samples.

**Indirect cause effect** is inevitable to be involved in the perturbation effect measurement. In a complex system, the causal effect may be continuously spread to more and more indirect targets as time goes forward, even going back to the cause at a later time. To avoid this kind of influence and focus on the relatively direct cause-effect relations, we limit the causal effect calculation to a short-time window, only considering the first half-clock after doing perturbation of a specific cause transistor. For example, if we perturb a transistor at half-clock 64, we calculate the causal effect by comparing the state sequences of regular runtime and perturbed runtime from half-clock 64 to 65.

Taking the aforementioned factors into account, we perform single-element perturbations on each unique transistor. We define a *period* as a recording of the entire microprocessor for a fixed length of $l$ half-clocks, where a half-clock is a runtime unit in the microprocessor. To address the challenge of learning with rare positive samples, we collect a total of 320 periods of microprocessor recordings and their corresponding perturbation recordings, ranging from half-clock 0 to 40960. Each period spans 128 half-clocks. We execute perturbations on each transistor at the midpoint of every period (Algorithm 1, line 4). For instance, for the period from half-clock 0 to 128, we separately perturb every transistor at half-clock 64; for the period from half-clock 128 to 256, we separately perturb every transistor at half-clock 192. We use $do(\boldsymbol{x} = p)$ to denote the perturbation operation, $p$ here represents the perturbation value (the value we intervene). We denote the first half-clock state segment of transistor $j$ when perturbing transistor $i$ at period $m$ after the perturbation as $\boldsymbol{x}_{j,do(\boldsymbol{x}_i=p),m}$. We denote the corresponding one half-clock state sequence of transistor $j$ when no transistor is perturbed as $\boldsymbol{x}_{j,m}$. It is important to note that the perturbed state sequence starts differently from the regular state sequence, with the perturbed state sequence commencing at the midpoint of the regular runtime. We define and calculate the temporal causal effect (TCE) of transistor $i$ on transistor $j$ when the perturbation occurs at period $m$ as follows (Algorithm 1, line 10):

$$TCE_{i,j,m} = E[|\boldsymbol{x}_{j,do(\boldsymbol{x}_i=p),m} - \boldsymbol{x}_{j,m}|] \tag{1}$$

We then convert TCEs into a binary adjacency matrix $\boldsymbol{A}$ to describe the relationship between any pair of transistors using the equation below. Transistor $i$ is considered the cause of transistor $j$ if the temporal causal effect $TCE_{i,j,m}$ is non-zero, while for reversed causal and unrelated relationships, the temporal causal effect should be zero (Algorithm 1, line 11). The adjacency matrix $\boldsymbol{A}_{i,j,m}$ is defined under the perturbation at half-clock $m$, necessitating the merging of recordings and adjacency matrices from different periods in the subsequent data preprocessing step. The zero threshold here is limited for systems with state values in floating-point numbers. For a system with states in floating-point values, we could use the average temporal causal effect across periods or multiple measurements and convert the classification task into a regression task.

$$\boldsymbol{A}_{i,j,m} = \mathbf{1}_{\text{TCE}_{i,j,m}>0} \tag{2}$$

---

**Algorithm 1** Adjacency Matrix Generation for M Periods

---

1: Start with $N$ observed elements inside system, $M$ periods
2: **for** $m = 0, 1, ..., M-1$ **do**
3:     **for** $i = 0, 1, ..., N-1$ **do**
4:         Perturb $element_i$ in period m and record all elements' states $\{\boldsymbol{x}_{m,0}, \boldsymbol{x}_{m,1}, ..., \boldsymbol{x}_{m,N}\}$
5:     **end for**
6: **end for**
7: **for** $m = 0, 1, ..., M-1$ **do**
8:     **for** $i = 0, 1, ..., N-1$ **do**
9:         **for** $j = 0, 1, ..., N-1$ **do**
10:             Calculate temporal causal effect $TCE_{i,j,m}$ from $element_i$ to $element_j$ in period $m$ as equation 1
11:             Calculate the entry $\boldsymbol{A}_{i,j,m}$ of adjacency matrix as equation 2
12:         **end for**
13:     **end for**
14: **end for**

---

## 2.2 Learning procedure

In this section, we outline the causal discovery procedure based on deep learning in two steps: data preprocessing and baseline setting. We focus on causal discovery in the game Donkey Kong and utilize simulation data obtained from the single-element perturbation experiment, which includes the regular runtime state

sequence $\boldsymbol{x}$ and the adjacency matrix $\boldsymbol{A}$. In the causal discovery procedure, we aim to explore the potential for inferring pairwise causality from observational data without direct access to any perturbation. As a result, the adjacency matrix $\boldsymbol{A}$ is only employed as a label for supervised learning and algorithm validation. Following data preprocessing, the processed state sequences of transistor pairs are fed into our discovery architecture. We assume that the mapping from observational data to causal relationships is stable and optimize the estimator through maximum likelihood estimation (Algorithm 2, lines 3-5).

---

**Algorithm 2** Learning Procedure

---

**Input:** training set $\mathbb{T} = \{(\boldsymbol{X}_n, \boldsymbol{A}_n)\}_{n=1}^{N_{data}}$, a dataset of pairwise element state sequences; sample $\boldsymbol{X}_n = (\boldsymbol{x}_{i,m} \parallel \boldsymbol{x}_{j,m})$, the stack of pairwise state sequences of $element_i$ and $element_j$ in the period $m$, $\boldsymbol{A}_n$ is the entry $\boldsymbol{A}_{i,j,m}$ of adjacency matrices.
**Input:** $\theta$, initial estimator parameters
**Output:** $\hat{\theta}$, the trained causality estimator
**Hyperparameters:** $N_{epochs} \in \mathbb{N}, \eta \in (0, \infty)$,
  1: **for** $i = 1, 2, ..., N_{epochs}$ **do**
  2:   **for** $n = 1, 2, ..., N_{data}$ **do**
  3:     $P \leftarrow Estimator(Causal | \boldsymbol{X}_n, \theta)$
  4:     $L(\theta) = -(\boldsymbol{A}_n \cdot \log(P) + (1 - \boldsymbol{A}_n) \cdot \log(1 - P))$
  5:     $\theta \leftarrow \theta - \eta \cdot \nabla L(\theta)$
  6:   **end for**
  7: **end for**
  8: **return** $\hat{\theta} = \theta$

---

### 2.2.1 Data preprocessing

To assess the ability to discover causality in another system at an arbitrary time, we validate our estimator on an unseen portion of transistors and unseen periods. For each period, we eliminate duplicates and use transistors belonging to the first 1755 transistors as training transistors, while employing those from the remaining 1755 transistors as test transistors. We remove several outlier periods with too few transistors after eliminating duplicates. Next, we use the test transistors in the first five periods ($0-128, 128-256, 384-512, 512-640, 640-768$) as five testing sets and the sixth period ($768-896$) as the validation set. Training transistors in the remaining periods serve as the training set. We then construct transistor pairs within each set independently, treating each pairwise relationship as a data sample. For example, although transistors $i$ and $j$, and transistors $j$ and $i$ constitute the same sequences, they are considered different data samples (Algorithm 2: Input). To mitigate the negative impact of sample imbalance, negative samples are randomly under-sampled to three times the quantity of positive samples in the training set. We maintain the validation and test sets unchanged to better reflect reality and evaluate the discovery quality of our procedure.

### 2.2.2 Architecture

We present the architecture for learning domain-specific causal discovery from time series. The high-resolution state recording is sparse and lengthy, which makes directly feeding it into the network computationally challenging. A closer inspection of our state sequences in the microprocessor system reveals numerous constant states and relatively fewer turning points (see Figure 5). Consequently, we require the model to discriminate and capture informative time windows where cause-effect relationships may be discernible. The system must also accommodate long input sequences. Inspired by Dosovitskiy et al. (2020), we use a similar architecture to Vision Transformer to handle temporal features. We first stack pairwise input into a vector, denoted by $\boldsymbol{X} \in \mathbb{R}^{L \times 2}$. Then we apply random shift augmentation to the input sequence $\boldsymbol{X}$ to enhance the estimator's generalization capabilities. We evenly crop $\boldsymbol{X}$ into windows of $\boldsymbol{X}$ and use a window embedding layer to project them to window embeddings $\boldsymbol{X}_w \in \mathbb{R}^{N \times C}$, where $C$ is the dimension of window embedding. We employ a 1D convolutional layer as the window embedding layer, as demonstrated in works like Text-CNN (Kim, 2014) and TCN (Lea et al., 2016), which proved the effectiveness and efficiency of convolutional kernels in extracting pattern information from sequences. We expect the window embedding layer to capture causal features within individual windows, such as the temporal lag of effect over a short

time. Additionally, we include a [*class*] token as the first window embedding and encode position information for each window. To capture information across windows and aggregate window features, the window embeddings are fed into a transformer (Vaswani et al., 2017). The transformer's output is globally pooled by an attention head to better weigh the importance of different time windows (Algorithm 2, lines 3-5). We thus use a relatively standard architecture from the time series or natural language field. See Appendix A.1.1 for the architecture and training details.

## 3    Empirical study

Intending to uncover the causal effect inside the MOS 6502 and the robustness of our learning causal discovery procedure, we carry out multiple empirical studies, including a causal effect analysis, regular evaluation, noise tests, generalization assessment on different games and explanation study in the context of MOS 6502. To test the generality of the approach, we then also test on the NetSim (Smith et al., 2011) dataset of simulated fMRI results, and the Dream3 (Prill et al., 2010) dataset to see how it works in a small sample case.

### 3.1    Causal effect

It is both an interesting and difficult question to determine how much causal influence there is and what role it plays in a complex system. Single element perturbation on each transistor provides us with the causal effect of every individual transistor. Figure 3 shows the temporal causal effect of three individual transistors, 134, 3355, and 452 when perturbing them at half-clock 960. Figure 4 shows the temporal causal effect of transistor 452 when perturbing it at different time points, specifically at half-clock 704, 832, and 960.

In Figure 3, the transistor 134, 3355, and 452 exhibit different temporal causal effects on the other transistors, which suggests variation in the causal relationship between the cause transistor and other transistors. We see that even though the connected wires guarantee that the voltage change is transferred to its following transistors, the perturbation of transistor 134 does not cause the change of any other transistors at that time (see Figure 3.A). This demonstrates that physical connection information might not be enough to provide complete information about causality. Moreover, compared to transistor 134 and 3355, transistor 452 caused much more transistors (see Figure 3.C). It is because the transistor 452 has an effect transistor that is directly connected to cclk wire, a wire connected to the clock register. The clock register is a module used to synchronize other modules of the microprocessor. It enables transistor 452 to indirectly have a causal effect on most transistors of the microprocessor. The strong causal effect of transistor 452 indicates its crucial role in the microprocessor. We see that the distribution of temporal causal effect is associated with functionality and importance. In addition, the effects of transistor 3355 and transistor 452 are different but similar in different games, which suggests the stable underlying causal structure across different behaviors (see Figure 3.B and C). The causal effect in the MOS 6502 system is nontrivial and strongly associated with different functional areas.

Interestingly and surprisingly, we find that the causal effect is associated with the perturbation time (see Figure 4). Here we present the causal effect of perturbing transistor 452 at different time points (half-clock 704, 832 and 960). Figure 4.A shows an interesting phenomenon that the causal effect of perturbing transistor 452 is changing across time, suggesting that the underlying causal structure of the microprocessor is changing across time. While transistor 452 causes a large number of transistors in half-clock 960 (see Figure 4.C), it only causes several transistors at half-clock 832 and barely any transistor at half-clock 832 (see Figure 4.A and B). We think it is partially due to the interaction of the Television Interface Adaptor (TIA) chip and microprocessor, which is like a hidden confounder to the whole microprocessor system (see Appendix A.2 for the analysis of causal pairs amount across time). It is also partially affected by the introduction of the MIMO structure of netlist and the indirect causal effect. Although this dynamic causal structure brings more challenges for causal discovery, we believe that the pattern of the pairwise causal relationship between transistors across time is stable and can be discovered. We thus examine causal discovery ability across different periods in the evaluation part as well. Having defined causality in the microprocessor thus sets us up to check how well we can do causal discovery from time series.

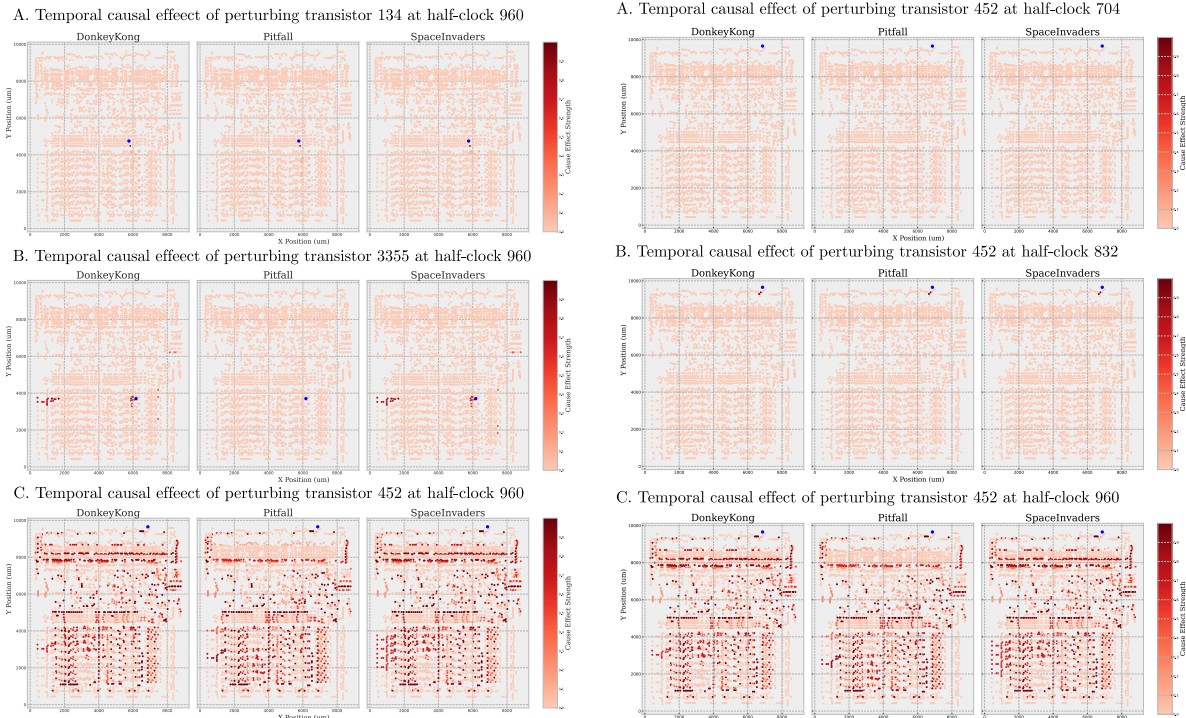

Figure 3: Causal effect of perturbations at the half-clock 960 inside MOS 6502 when executing different games. Panels are MOS 6502 board, and small round units on it are transistors. The blue round unit is the perturbed transistor and the red round units are the other transistors caused by the perturbed transistor, with different shades of red representing the strength of the causal effect. X and Y positions indicate the horizontal and vertical position of a transistor on the microprocessor board. **A.** Temporal causal effect (TCE, measured during one half-clock, 30 time steps) when stimulating transistor 1; **B.** TCE of transistor 3355; **C.** TCE of transistor 452

Figure 4: Causal effect of perturbations at three different time points inside MOS 6502 when executing different games (Figure settings are same as Figure 3). **A.** Temporal causal effect (TCE) when perturbing transistor 452 at half-clock 704; **B.** TCE of perturbing transistor 452 at half-clock 832; **C.** TCE of perturbing transistor 452 at half-clock 960

## 3.2 Evaluation of classical and learned causal discovery processes

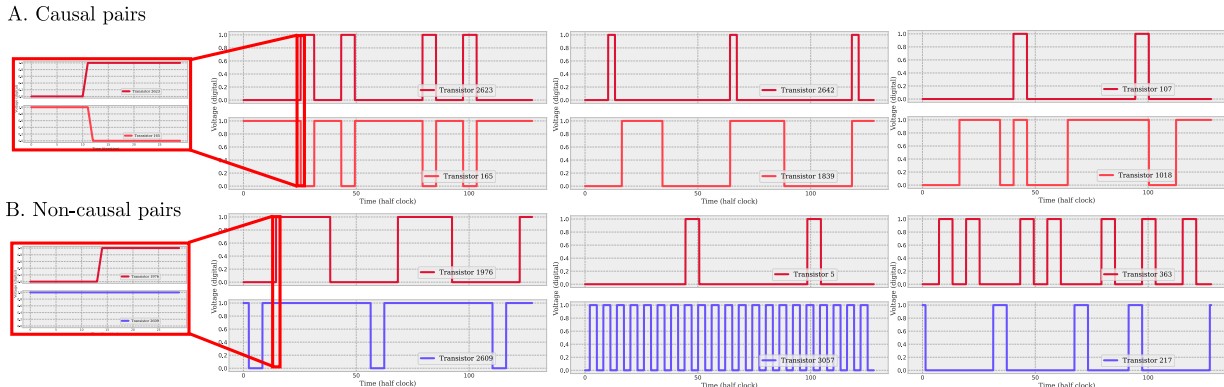

Figure 5: Transistor pair examples of different relationships. **A.** Causal pairs of transistors: 2623 *to* 165, 2642 *to* 1839 and 107 *to* 1018. **B.** Non-causal pairs of transistors: 1976 *and* 2609, 5 *and* 3057, and 363 *and* 217

Many aspects of time series may be indicative of causality. The left part of Figure 5 shows some examples of causal and non-causal pairs in the period from half-clock 896 to half-clock 1024, uncovered by perturbation analysis. Taking a zoom-in view of transistors 2623 and 165 shows that the causal feature here is the time lag between cause and effect in the turning point of cause. Although it is relatively easy to recognize the first causal pair, more various difficult scenarios like the other two positive samples are beyond human observation and assumptions, which might have potential specialized causal features in a specific kind of complex system. It is unclear if the causal discovery methods that are starting with human assumptions and are designed for general use, can still work well on more complex realistic cases. After all, these algorithms do not consider the special features of a specific domain's system and the algorithm designers may be unable to appreciate those features. Many features may be usable to infer causality from the time series.

Here we evaluate how well our learned estimator does and compare them with traditional bi-variate relativity metrics: Pearson Correlation (Corr), Mutual Information (MI), and Linear Granger Causality (GC) (Granger, 1969), VAR-LiNGAM (LiNGAM) (Shimizu et al., 2006). Due to the large amount of test data and the size of the causal graph, we choose relatively fast pair-wise methods to compare. We use the area under the receiver operating characteristic (AUROC) and the area under the precision-recall curve (AUPRC, also called average precision) to evaluate how these methods work. We test all methods on pairs of time series, just like when training our models. We can thus compare the learned estimator and classical techniques (see Appendix A.1.2 for experiment details).

Table 1: Models' performance on different games and periods.

| Game | Period | AUROC | | | | | AUPRC | | | | |
|---|---|---|---|---|---|---|---|---|---|---|---|
| | | Corr | MI | LiNGAM | GC | Transformer | Corr | MI | LiNGAM | GC | Transformer |
| Donkey Kong | [0, 128] | 0.98 | 0.95 | 0.89 | 0.83 | **0.98** | 0.18 | 0.15 | 0.10 | 0.25 | **0.55** |
| | [128, 256] | 0.88 | 0.88 | 0.90 | 0.87 | **1.00** | 0.12 | 0.16 | 0.06 | 0.15 | **0.49** |
| | [384, 512] | 0.92 | 0.90 | 0.91 | 0.90 | **0.96** | 0.12 | 0.09 | 0.09 | 0.10 | **0.41** |
| | [512, 640] | 0.62 | 0.64 | 0.65 | 0.58 | **0.93** | 0.05 | 0.05 | 0.05 | 0.04 | **0.33** |
| | [640, 768] | 0.94 | 0.94 | 0.91 | 0.89 | **1.00** | 0.19 | 0.19 | 0.10 | 0.12 | **0.60** |
| | $mean \pm std$ | $0.87 \pm 0.13$ | $0.86 \pm 0.11$ | $0.85 \pm 0.10$ | $0.81 \pm 0.12$ | **0.97±0.03** | $0.13 \pm 0.05$ | $0.13 \pm 0.05$ | $0.08 \pm 0.02$ | $0.13 \pm 0.07$ | **0.48±0.10** |
| Pitfall | [0, 128] | 0.98 | 0.96 | 0.88 | 0.83 | **0.99** | 0.15 | 0.10 | 0.02 | 0.21 | **0.49** |
| | [128, 256] | 0.88 | 0.87 | 0.89 | 0.88 | **0.99** | 0.11 | 0.14 | 0.06 | 0.19 | **0.49** |
| | [384, 512] | 0.92 | 0.89 | 0.89 | 0.82 | **0.95** | 0.07 | 0.06 | 0.03 | 0.07 | **0.44** |
| | [512, 640] | 0.60 | 0.62 | 0.66 | 0.63 | **0.91** | 0.05 | 0.06 | 0.04 | 0.05 | **0.28** |
| | [640, 768] | 0.92 | 0.91 | 0.93 | 0.83 | **0.98** | 0.17 | 0.17 | 0.05 | 0.14 | **0.49** |
| | $mean \pm std$ | $0.86 \pm 0.13$ | $0.85 \pm 0.12$ | $0.85 \pm 0.10$ | $0.80 \pm 0.09$ | **0.96±0.03** | $0.11 \pm 0.05$ | $0.11 \pm 0.04$ | $0.04 \pm 0.01$ | $0.13 \pm 0.06$ | **0.44±0.08** |
| Space Invaders | [0, 128] | 0.98 | 0.95 | 0.92 | 0.83 | **0.99** | 0.15 | 0.12 | 0.03 | 0.19 | **0.50** |
| | [128, 256] | 0.85 | 0.85 | 0.91 | 0.84 | **0.98** | 0.07 | 0.09 | 0.04 | 0.12 | **0.50** |
| | [384, 512] | 0.88 | 0.86 | 0.87 | 0.86 | **0.96** | 0.07 | 0.07 | 0.06 | 0.11 | **0.56** |
| | [512, 640] | 0.62 | 0.64 | 0.66 | 0.59 | **0.89** | 0.04 | 0.03 | 0.04 | 0.05 | **0.25** |
| | [640, 768] | 0.93 | 0.93 | 0.88 | 0.84 | **0.98** | 0.17 | 0.19 | 0.13 | 0.14 | **0.64** |
| | $mean \pm std$ | $0.85 \pm 0.12$ | $0.85 \pm 0.11$ | $0.85 \pm 0.10$ | $0.79 \pm 0.10$ | **0.96±0.04** | $0.10 \pm 0.05$ | $0.10 \pm 0.05$ | $0.06 \pm 0.04$ | $0.12 \pm 0.05$ | **0.49±0.13** |

We use all the methods using data from the game Donkey Kong and will only use other games for testing procedures (see Table 1). Interestingly, we find AUROC is not always suitable for causal discovery evaluation, since the causal components in the real world are usually rare and AUROC is not that sensitive to the minority positive class. For example, in an extremely imbalanced dataset such as MOS 6502, misclassifying most of the minority positive class can still lead to a good ROC curve. Therefore, considering the ability to correctly recognize causality (positive class) as one of the most critical abilities in causal discovery, we also report AUPRC here, which cares more about how well a discovery method recognizes causality. Here we execute two different games, Pitfall and Space Invaders on the MOS 6502 and regard them as different behaviors executed by the same system. Note that the estimator is learned only on Donkey Kong and directly tested on Pitfall and Space Invaders.

Learned estimator far outperforms the methods based on human intuition in the game Donkey Kong (see the first block of Table 1). For the AUROC, the learned estimator achieve the best performance with almost perfect results in two periods. While methods based on human intuition also have good and close performance to the learned estimator on most of periods, they perform much worse than the learned estimator on the specific period $[512, 640]$, where positive samples are much more than the other periods. For the AUPRC, the learned estimator achieves an average $\sim 0.5$ while traditional methods get only $\sim 0.2$. It shows that these human assumption based methods designed for general use are not able to predict what causality is in specific domains. Benefiting from the supervision signal of perturbation and a big data-driven way, our learned estimator can discover what causality is in the microprocessor system and use it to infer causality in the unseen part and time. Especially on the period $[512, 640]$ where traditional methods degenerate a lot, our learned estimator exhibits stable and superior generalization. As a semantically similar concept to causality and a common tool to explore connectivity, the correlation here achieves only $\sim 0.1$ AUPRC. Clearly, the learned domain-specific causal estimator far outperforms the methods based on human intuition.

Learned estimator can be generalized to different behaviors conducted on the same domain system (see the second and third blocks of Table 1). It shows that despite that our estimator is not learned from these games, it can outperform traditional methods both for AUROC and AUPRC and even show better performance on some periods than Donkey Kong. We find very strong generalizations of the learned estimator across games and across times in the same domain, which is the result of our model having learned the causal dynamics among transistors.

### 3.3 Evaluation under noise

In contrast to ideal microprocessor simulation, real-world causal discovery often happens in domains where there are all kinds of noise. Independent from the system itself, observational noise is often common in real-world scenarios. For instance, the wire used to sample the voltages of neurons might be perturbed by the wind while recording brain waves (wind brings small observational noise to the brain wave signal recorded). There is a lot of tool/observation noise other than noise in the system, collected with data together. Intending to explore the potential effectiveness of our methods in other real-world scenarios, we simulate the small-scale observational noise by simply adding Gaussian noise of different scales in the testing data. Here we use the estimator learned from the noise-free Donkey Kong training set to infer the causality in the noised test set. We scale the standard normal distribution with factor $0.03, 0.05, 0.1$ and add it to the noise-free test set. Our method still has stable and good AUROC and outperforms traditional methods (see Table 2). While the AUPRC of all methods is affected by the additive observational noise, our learned estimator does not degenerate a lot and still has a much stronger ability to discover causality than the other methods. Especially VAR-LiNGAM, which assumes non-gaussian noise, degenerates the most when introducing Gaussian noise. This benefits from the Transformer's feature exacting ability and introduced supervised signal, which, arguably, makes our model still capable of uncovering causality under small-scale noise and outperforms other methods based on human intuition. We also explore our learned estimator's ability under large scales of noise such as 0.3 and 0.5 (see Appendix A.6.1), finding its limitation under large-scale observational noise.

Table 2: Models' performance on Donkey Kong under the noise of different scales.

| Noise Scale | Period | AUROC | | | | | AUPRC | | | | |
|---|---|---|---|---|---|---|---|---|---|---|---|
| | | Corr | MI | LiNGAM | GC | Transformer | Corr | MI | LiNGAM | GC | Transformer |
| 0.03 | [0, 128] | 0.98 | 0.95 | 0.94 | 0.98 | **0.98** | 0.17 | 0.15 | 0.21 | 0.25 | **0.55** |
| | [128, 256] | 0.88 | 0.88 | 0.85 | 0.85 | **1.00** | 0.13 | 0.16 | 0.19 | 0.09 | **0.48** |
| | [384, 512] | 0.92 | 0.90 | 0.92 | 0.85 | **0.96** | 0.13 | 0.11 | 0.17 | 0.09 | **0.41** |
| | [512, 640] | 0.62 | 0.64 | 0.57 | 0.46 | **0.93** | 0.05 | 0.05 | 0.06 | 0.03 | **0.33** |
| | [640, 768] | 0.94 | 0.94 | 0.95 | 0.75 | **1.00** | 0.18 | 0.19 | 0.29 | 0.09 | **0.60** |
| | $mean \pm std$ | $0.87 \pm 0.13$ | $0.86 \pm 0.11$ | $0.85 \pm 0.14$ | $0.78 \pm 0.17$ | **0.97±0.03** | $0.13 \pm 0.05$ | $0.13 \pm 0.05$ | $0.18 \pm 0.07$ | $0.11 \pm 0.07$ | **0.47±0.10** |
| 0.05 | [0, 128] | 0.98 | 0.96 | 0.92 | 0.94 | **0.98** | 0.18 | 0.11 | 0.12 | 0.26 | **0.52** |
| | [128, 256] | 0.88 | 0.89 | 0.88 | 0.84 | **1.00** | 0.14 | 0.16 | 0.11 | 0.12 | **0.47** |
| | [384, 512] | 0.92 | 0.89 | 0.92 | 0.92 | **0.96** | 0.14 | 0.08 | 0.11 | 0.13 | **0.38** |
| | [512, 640] | 0.62 | 0.63 | 0.58 | 0.51 | **0.93** | 0.05 | 0.05 | 0.05 | 0.03 | **0.33** |
| | [640, 768] | 0.94 | 0.93 | 0.96 | 0.91 | **1.00** | 0.19 | 0.16 | 0.30 | 0.11 | **0.56** |
| | $mean \pm std$ | $0.87 \pm 0.13$ | $0.86 \pm 0.12$ | $0.85 \pm 0.14$ | $0.82 \pm 0.16$ | **0.97±0.03** | $0.14 \pm 0.05$ | $0.11 \pm 0.04$ | $0.14 \pm 0.08$ | $0.13 \pm 0.07$ | **0.45±0.09** |
| 0.1 | [0, 128] | 0.97 | 0.96 | 0.90 | 0.98 | **0.98** | 0.15 | 0.14 | 0.07 | 0.23 | **0.47** |
| | [128, 256] | 0.89 | 0.88 | 0.86 | 0.91 | **1.00** | 0.14 | 0.15 | 0.04 | 0.13 | **0.34** |
| | [384, 512] | 0.92 | 0.89 | 0.88 | 0.93 | **0.96** | 0.12 | 0.09 | 0.08 | 0.13 | **0.40** |
| | [512, 640] | 0.63 | 0.63 | 0.61 | 0.60 | **0.93** | 0.05 | 0.05 | 0.03 | 0.04 | **0.31** |
| | [640, 768] | 0.94 | 0.93 | 0.96 | 0.95 | **0.99** | 0.18 | 0.20 | 0.12 | 0.11 | **0.48** |
| | $mean \pm std$ | $0.87 \pm 0.12$ | $0.86 \pm 0.12$ | $0.84 \pm 0.12$ | $0.87 \pm 0.14$ | **0.97±0.02** | $0.13 \pm 0.04$ | $0.13 \pm 0.05$ | $0.07 \pm 0.03$ | $0.13 \pm 0.06$ | **0.40±0.07** |

### 3.4 Test if the methods also work in a different domain of causal discovery, fMRI data

Sudden and gradual changes both happen in real-world scenarios. While experiments in MOS 6502 focus on the sudden changes in digital voltage, we here examine our procedure in NetSim (Smith et al., 2011), a synthesized fMRI dataset modeling gradual changes in brain networks (used as a benchmark in Löwe et al. (2022); Khanna & Tan (2019); Marinescu et al. (2018); Gong et al. (2022)). There are in total 28 simulations generated with different specified system properties such as external input strength and the number of nodes in the system. Each simulation has a fixed number of nodes $N$ and a specified property in all samples, where a sample refers to $N$ state sequences of $N$ nodes and its corresponding $N \times N$ connection matrix. It is a classical causal discovery setting allowing us to compare the multi-variate and graph-based discovery methods with our proposed approach.

We choose 17 simulations that have the same sequence length (200). We use the first 60% subjects of the simulation $1, 2, 3, 4, 8$ as the training set and use the last 20% subjects of each simulation as the test set. We omit perturbations here (we have the full ground truth connectivity matrix after all) and directly convert all the connection strengths into a binary coding of connections so that they could be represented as causal relationships. Here we report the mean AUROC and AUPRC across subjects in each simulation and the standard deviation of them. We skip the evaluation on the simulation 4 because it is a large graph and adopting graph-based methods is very time-consuming on it. We test human assumption based time-series causal discovery methods including pair-wise based and graph-based methods such as Pearson Correlation (Corr), DYNOTEARS (DYNO) (Pamfil et al., 2020), Multivariate Granger Causality (GC) Shojaie & Michailidis (2010), Mutual Information (MI), PCMCI+ (Peters et al., 2013), VAR-LiNGAM (LiNGAM) (Hyvärinen et al., 2010), Neural Granger Causality (cMLP, cLSTM) (Tank et al., 2021), eSRU and SRU (Khanna & Tan, 2019). We see that in Table 13 (see Appendix A.4) and Table 3, only learning from the distributions of simulation $1, 2, 3, 4, 8$, our method still outperforms on most of the simulations and achieve near perfect result on several simulations, especially on the AUPRC evaluation. While other methods reach below 0.5 AUPRC, the learned estimator achieves $\sim 0.9$ in most cases. It shows the importance of learning causality with perturbation information. However, our procedure is suboptimal on the simulation $22, 23, 24$, since they have very different distribution and our estimator have not learned such dynamics. In most cases, learned causal discovery works very well (see Appendix A.1.3 for experiment details and Appendix A.4 for inferred adjacency matrix examples).

Here we evaluate the ability to tolerate small-scale observational noise on NetSim as we have done in subsection 3.3. We simulate such noise as we have done in the MOS 6502 in subsection 3.3. Note that we also directly test the network we get from the noise-free training set on the test set with additive noise of scale factor 0.1. The additive noise is normalized to the same scale of the raw data of each subject. Although adding noise affects all the methods including the learned estimator, the learned estimator performance does not decrease a lot and still outperforms other methods significantly on both AUROC and AUPRC (see Table 7 and 4). Our method presents consistent noise-tolerate ability as it does in the MOS 6502. Supervised by

Table 3: AUPRC comparison on different simulations of NetSim (mean ± std).

| Dataset | Corr | DYNO | GC | MI | PCMCI+ | LiNGAM | cMLP | cLSTM | eSRU | SRU | Transformer |
|---|---|---|---|---|---|---|---|---|---|---|---|
| Sim1 | $0.47 \pm 0.02$ | $0.41 \pm 0.08$ | $0.40 \pm 0.08$ | $0.39 \pm 0.08$ | $0.39 \pm 0.09$ | $0.43 \pm 0.15$ | $0.42 \pm 0.15$ | $0.41 \pm 0.14$ | $0.40 \pm 0.14$ | $0.39 \pm 0.14$ | **0.97±0.06** |
| Sim2 | $0.45 \pm 0.03$ | $0.33 \pm 0.12$ | $0.32 \pm 0.12$ | $0.31 \pm 0.11$ | $0.29 \pm 0.11$ | $0.30 \pm 0.11$ | $0.29 \pm 0.11$ | $0.28 \pm 0.11$ | $0.27 \pm 0.11$ | $0.26 \pm 0.10$ | **0.94±0.05** |
| Sim3 | $0.44 \pm 0.03$ | $0.32 \pm 0.13$ | $0.29 \pm 0.14$ | $0.27 \pm 0.12$ | $0.26 \pm 0.12$ | $0.27 \pm 0.13$ | $0.26 \pm 0.12$ | $0.24 \pm 0.12$ | $0.23 \pm 0.12$ | $0.22 \pm 0.12$ | **0.89±0.06** |
| Sim8 | $0.41 \pm 0.07$ | $0.36 \pm 0.08$ | $0.38 \pm 0.11$ | $0.37 \pm 0.10$ | $0.36 \pm 0.10$ | $0.40 \pm 0.14$ | $0.40 \pm 0.14$ | $0.39 \pm 0.14$ | $0.39 \pm 0.14$ | $0.38 \pm 0.13$ | **0.85±0.13** |
| Sim10 | $0.48 \pm 0.02$ | $0.38 \pm 0.10$ | $0.39 \pm 0.12$ | $0.40 \pm 0.11$ | $0.40 \pm 0.12$ | $0.42 \pm 0.16$ | $0.42 \pm 0.16$ | $0.42 \pm 0.15$ | $0.42 \pm 0.15$ | $0.42 \pm 0.15$ | **0.96±0.03** |
| Sim11 | $0.28 \pm 0.03$ | $0.26 \pm 0.04$ | $0.26 \pm 0.06$ | $0.26 \pm 0.06$ | $0.25 \pm 0.07$ | $0.25 \pm 0.08$ | $0.25 \pm 0.08$ | $0.24 \pm 0.08$ | $0.24 \pm 0.08$ | $0.23 \pm 0.08$ | **0.71±0.10** |
| Sim12 | $0.43 \pm 0.02$ | $0.36 \pm 0.08$ | $0.33 \pm 0.11$ | $0.31 \pm 0.10$ | $0.29 \pm 0.11$ | $0.30 \pm 0.11$ | $0.28 \pm 0.11$ | $0.27 \pm 0.11$ | $0.26 \pm 0.11$ | $0.26 \pm 0.11$ | **0.90±0.06** |
| Sim13 | $0.48 \pm 0.04$ | $0.47 \pm 0.05$ | $0.48 \pm 0.07$ | $0.49 \pm 0.10$ | $0.47 \pm 0.10$ | $0.48 \pm 0.10$ | $0.47 \pm 0.10$ | $0.47 \pm 0.10$ | $0.47 \pm 0.11$ | $0.46 \pm 0.11$ | **0.76±0.10** |
| Sim14 | $0.48 \pm 0.02$ | $0.41 \pm 0.08$ | $0.41 \pm 0.09$ | $0.40 \pm 0.08$ | $0.38 \pm 0.09$ | $0.42 \pm 0.13$ | $0.41 \pm 0.13$ | $0.40 \pm 0.13$ | $0.39 \pm 0.13$ | $0.39 \pm 0.13$ | **0.93±0.08** |
| Sim15 | $0.45 \pm 0.03$ | $0.38 \pm 0.07$ | $0.40 \pm 0.09$ | $0.41 \pm 0.08$ | $0.41 \pm 0.10$ | $0.48 \pm 0.21$ | $0.47 \pm 0.20$ | $0.45 \pm 0.20$ | $0.44 \pm 0.19$ | $0.43 \pm 0.19$ | **0.72±0.09** |
| Sim16 | $0.48 \pm 0.01$ | $0.44 \pm 0.05$ | $0.45 \pm 0.07$ | $0.44 \pm 0.06$ | $0.44 \pm 0.06$ | $0.46 \pm 0.10$ | $0.46 \pm 0.10$ | $0.45 \pm 0.10$ | $0.45 \pm 0.10$ | $0.45 \pm 0.10$ | **0.96±0.03** |
| Sim17 | $0.47 \pm 0.01$ | $0.39 \pm 0.09$ | $0.36 \pm 0.10$ | $0.36 \pm 0.09$ | $0.35 \pm 0.10$ | $0.42 \pm 0.19$ | $0.40 \pm 0.19$ | $0.37 \pm 0.19$ | $0.35 \pm 0.19$ | $0.34 \pm 0.19$ | **0.98±0.02** |
| Sim18 | $0.48 \pm 0.03$ | $0.42 \pm 0.07$ | $0.42 \pm 0.12$ | $0.41 \pm 0.10$ | $0.40 \pm 0.11$ | $0.43 \pm 0.16$ | $0.42 \pm 0.16$ | $0.41 \pm 0.16$ | $0.40 \pm 0.15$ | $0.39 \pm 0.15$ | **0.99±0.02** |
| Sim21 | $0.48 \pm 0.03$ | $0.42 \pm 0.08$ | $0.41 \pm 0.08$ | $0.40 \pm 0.08$ | $0.38 \pm 0.09$ | $0.42 \pm 0.15$ | $0.41 \pm 0.14$ | $0.40 \pm 0.14$ | $0.39 \pm 0.14$ | $0.38 \pm 0.13$ | **0.95±0.06** |
| Sim22 | **0.41±0.04** | $0.38 \pm 0.06$ | $0.38 \pm 0.08$ | $0.39 \pm 0.07$ | $0.37 \pm 0.08$ | $0.37 \pm 0.09$ | $0.35 \pm 0.09$ | $0.34 \pm 0.09$ | $0.34 \pm 0.09$ | $0.34 \pm 0.09$ | $0.31 \pm 0.11$ |
| Sim23 | $0.40 \pm 0.04$ | $0.35 \pm 0.06$ | $0.40 \pm 0.12$ | $0.39 \pm 0.10$ | $0.41 \pm 0.14$ | **0.47±0.21** | $0.45 \pm 0.20$ | $0.43 \pm 0.19$ | $0.42 \pm 0.19$ | $0.41 \pm 0.19$ | $0.41 \pm 0.05$ |
| Sim24 | $0.36 \pm 0.06$ | $0.31 \pm 0.07$ | $0.34 \pm 0.10$ | $0.35 \pm 0.10$ | $0.35 \pm 0.11$ | $0.35 \pm 0.11$ | $0.34 \pm 0.11$ | $0.34 \pm 0.11$ | $0.34 \pm 0.11$ | $0.33 \pm 0.11$ | **0.37±0.11** |

causal ground truth help the estimator understand causality thus being noise tolerant. We also explore our learned estimator's limitation under large scales of noise such as 0.3 and 0.5 (see Appendix A.6.2), and find its instability to large-scale observational noise.

Table 4: AUPRC comparison on NetSim under noise with scale 0.1 (mean ± std).

| Dataset | Corr | DYNO | GC | MI | PCMCI+ | LiNGAM | cMLP | cLSTM | eSRU | SRU | Transformer |
|---|---|---|---|---|---|---|---|---|---|---|---|
| Sim1 | $0.43 \pm 0.05$ | $0.36 \pm 0.07$ | $0.39 \pm 0.13$ | $0.39 \pm 0.12$ | $0.37 \pm 0.11$ | $0.40 \pm 0.14$ | $0.40 \pm 0.14$ | $0.39 \pm 0.14$ | $0.39 \pm 0.14$ | $0.38 \pm 0.14$ | **0.78±0.19** |
| Sim2 | $0.35 \pm 0.04$ | $0.25 \pm 0.10$ | $0.26 \pm 0.09$ | $0.24 \pm 0.09$ | $0.22 \pm 0.09$ | $0.24 \pm 0.10$ | $0.23 \pm 0.10$ | $0.23 \pm 0.09$ | $0.22 \pm 0.09$ | $0.22 \pm 0.09$ | **0.68±0.12** |
| Sim3 | $0.34 \pm 0.07$ | $0.23 \pm 0.12$ | $0.20 \pm 0.11$ | $0.18 \pm 0.10$ | $0.17 \pm 0.10$ | $0.18 \pm 0.10$ | $0.17 \pm 0.09$ | $0.17 \pm 0.09$ | $0.16 \pm 0.09$ | $0.16 \pm 0.08$ | **0.50±0.16** |
| Sim8 | $0.38 \pm 0.05$ | $0.33 \pm 0.07$ | $0.38 \pm 0.13$ | $0.37 \pm 0.12$ | $0.35 \pm 0.12$ | $0.38 \pm 0.14$ | $0.38 \pm 0.14$ | $0.39 \pm 0.14$ | $0.38 \pm 0.14$ | $0.38 \pm 0.14$ | **0.68±0.20** |
| Sim10 | $0.44 \pm 0.05$ | $0.35 \pm 0.10$ | $0.40 \pm 0.15$ | $0.39 \pm 0.14$ | $0.38 \pm 0.13$ | $0.40 \pm 0.14$ | $0.40 \pm 0.14$ | $0.40 \pm 0.14$ | $0.40 \pm 0.14$ | $0.38 \pm 0.14$ | **0.73±0.16** |
| Sim11 | $0.25 \pm 0.02$ | $0.21 \pm 0.04$ | $0.23 \pm 0.06$ | $0.21 \pm 0.07$ | $0.20 \pm 0.07$ | $0.21 \pm 0.07$ | $0.21 \pm 0.07$ | $0.21 \pm 0.07$ | $0.20 \pm 0.07$ | $0.20 \pm 0.07$ | **0.42±0.09** |
| Sim12 | $0.33 \pm 0.05$ | $0.26 \pm 0.09$ | $0.24 \pm 0.08$ | $0.22 \pm 0.08$ | $0.21 \pm 0.08$ | $0.22 \pm 0.09$ | $0.22 \pm 0.09$ | $0.21 \pm 0.08$ | $0.21 \pm 0.08$ | $0.20 \pm 0.08$ | **0.60±0.13** |
| Sim13 | $0.43 \pm 0.05$ | $0.41 \pm 0.06$ | $0.46 \pm 0.12$ | $0.45 \pm 0.12$ | $0.44 \pm 0.12$ | $0.47 \pm 0.14$ | $0.47 \pm 0.13$ | $0.47 \pm 0.13$ | $0.46 \pm 0.13$ | $0.45 \pm 0.13$ | **0.56±0.12** |
| Sim14 | $0.45 \pm 0.04$ | $0.37 \pm 0.09$ | $0.38 \pm 0.10$ | $0.36 \pm 0.10$ | $0.34 \pm 0.10$ | $0.37 \pm 0.13$ | $0.36 \pm 0.12$ | $0.36 \pm 0.12$ | $0.36 \pm 0.12$ | $0.35 \pm 0.12$ | **0.82±0.16** |
| Sim15 | $0.41 \pm 0.03$ | $0.34 \pm 0.08$ | $0.40 \pm 0.15$ | $0.40 \pm 0.14$ | $0.39 \pm 0.13$ | $0.42 \pm 0.15$ | $0.41 \pm 0.15$ | $0.41 \pm 0.14$ | $0.40 \pm 0.14$ | $0.39 \pm 0.14$ | **0.63±0.17** |
| Sim16 | $0.45 \pm 0.03$ | $0.41 \pm 0.05$ | $0.43 \pm 0.09$ | $0.42 \pm 0.08$ | $0.42 \pm 0.08$ | $0.43 \pm 0.10$ | $0.43 \pm 0.10$ | $0.44 \pm 0.10$ | $0.43 \pm 0.10$ | $0.43 \pm 0.10$ | **0.81±0.09** |
| Sim17 | $0.42 \pm 0.03$ | $0.31 \pm 0.12$ | $0.30 \pm 0.10$ | $0.27 \pm 0.10$ | $0.26 \pm 0.10$ | $0.28 \pm 0.12$ | $0.28 \pm 0.12$ | $0.26 \pm 0.12$ | $0.26 \pm 0.11$ | $0.25 \pm 0.11$ | **0.87±0.08** |
| Sim18 | $0.45 \pm 0.04$ | $0.37 \pm 0.09$ | $0.39 \pm 0.10$ | $0.37 \pm 0.10$ | $0.35 \pm 0.10$ | $0.40 \pm 0.14$ | $0.39 \pm 0.14$ | $0.39 \pm 0.14$ | $0.38 \pm 0.14$ | $0.37 \pm 0.13$ | **0.81±0.19** |
| Sim21 | $0.42 \pm 0.05$ | $0.36 \pm 0.07$ | $0.40 \pm 0.13$ | $0.38 \pm 0.12$ | $0.36 \pm 0.12$ | $0.38 \pm 0.14$ | $0.38 \pm 0.14$ | $0.38 \pm 0.15$ | $0.37 \pm 0.14$ | $0.37 \pm 0.14$ | **0.78±0.11** |
| Sim22 | **0.38±0.04** | $0.36 \pm 0.06$ | $0.38 \pm 0.10$ | $0.37 \pm 0.09$ | $0.36 \pm 0.10$ | $0.36 \pm 0.10$ | $0.36 \pm 0.10$ | $0.35 \pm 0.10$ | $0.35 \pm 0.10$ | $0.34 \pm 0.10$ | $0.36 \pm 0.14$ |
| Sim23 | $0.34 \pm 0.05$ | $0.30 \pm 0.06$ | $0.35 \pm 0.13$ | $0.34 \pm 0.12$ | $0.35 \pm 0.12$ | $0.37 \pm 0.14$ | $0.36 \pm 0.14$ | $0.36 \pm 0.13$ | $0.35 \pm 0.13$ | $0.35 \pm 0.13$ | **0.47±0.14** |
| Sim24 | $0.30 \pm 0.03$ | $0.28 \pm 0.03$ | $0.31 \pm 0.11$ | $0.31 \pm 0.10$ | $0.30 \pm 0.10$ | $0.31 \pm 0.10$ | $0.31 \pm 0.10$ | $0.31 \pm 0.10$ | $0.31 \pm 0.09$ | $0.30 \pm 0.09$ | **0.34±0.08** |

### 3.5 Test if the methods also work in a small sample case, gene network

Machine learning often fails on small sample cases, where models often suffer from sample bias and weak generalization. Interestingly, we find a similar issue in framing learning domain-specific causal discovery to a machine learning task. In the MOS 6502 experiments, we see that using only one period as the training set for learning gives good AUROC but only $\sim 0.25$ AUPRC. Here we test our method on another causal discovery dataset, Dream3 (Prill et al., 2010). Dream3 is a silico gene network benchmark. We choose five different graphs from it: Ecoli1, Ecoli2, Yeast1, Yeast2, and Yeast3. Each network consisted of 100 nodes and corresponding recordings with a time length of 966. Each gene recording contains 46 snippets of perturbation trajectories whose time length is 21. We use the same methods as NetSim except for VAR-LiNGAM to do the comparison, since the time length is too short for a 100 variates regression. We use Ecoli2, Yeast2, and Yeast3 as test sets and learn one estimator from Ecoli1 and Yeast1. For the learned estimator, we directly learn it from the full recording and test on the full recording, while testing other methods on the snippets and merging their result to a single graph (Experiment details can be found in Appendix A.1.4). Although our learned estimator still outperforms the other methods on most datasets, we find that our learned estimator presents unsatisfactory results on AUROC and AUPRC (see Table 5 and Table 8 in Appendix A.5). We see from the low AUPRC that due to extremely limited positive samples (average 270 positive samples compared to 10000 total samples per dataset), our estimator is biased toward the negative sample side and thus hard to learn the domain-specific causality, resulting in the problem of generalizing on the unseen test set. We also

see from the optimization process that it presents overfitting from the early stage. It shows that the small sample issue is crucial for learning domain-specific causal discovery since it is not enough for the estimator to learn what causality is in a certain domain, resulting in an ineffective learning process.

Table 5: AUPRC comparison on the Dream3

| Dataset | Corr | DYNO | GC | MI | PCMCI+ | cMLP | cLSTM | eSRU | SRU | Transformer |
|---|---|---|---|---|---|---|---|---|---|---|
| Ecoli2 | 0.01 | 0.01 | 0.01 | 0.02 | 0.01 | 0.01 | 0.02 | 0.01 | 0.01 | **0.04** |
| Yeast2 | 0.01 | 0.01 | 0.02 | 0.01 | 0.01 | 0.04 | 0.04 | 0.04 | 0.04 | **0.06** |
| Yeast3 | 0.01 | 0.01 | 0.01 | 0.01 | 0.01 | 0.06 | **0.07** | 0.06 | 0.06 | 0.06 |

### 3.6 Explanation study

We see that our learned estimator has shown good performance in causal discovery, but what if it is exploiting some shortcut that we humans do not understand and that is specific to the MOS 6502 or NetSim? To find out what our estimator has learned from the supervised signal, we use the estimator to conduct an explanation study. We first use Gradient-weighted Class Activation Mapping (Grad-CAM) (Selvaraju et al., 2016) to try to understand what the estimator learns. Then we try to inspect what happens when we violate the temporal order of cause-result by adding a small time shift to the cause transistor.

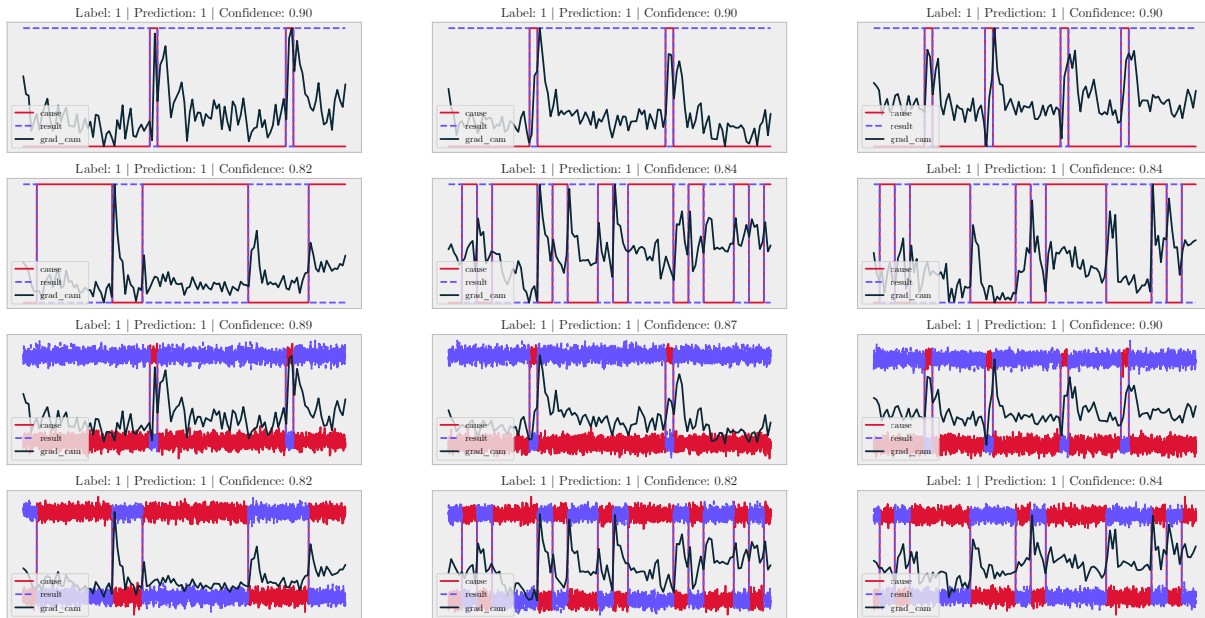

Figure 6: Grad-CAM mappings of causal pairs. Confidence refers to the estimation confidence of the positive class (causality). **A.** The gradient saliency mapping of three causal transistor pairs in the ideal context. **B.** The gradient saliency mapping of the same causal transistor pairs at a noise scale of 0.1.

Grad-CAM (Selvaraju et al., 2016) is a robust tool making Convolutional Neural Networks more transparent by helping localize saliency in the feature map. We adopt Grad-CAM mainly on the output of the attention block in the last layer and interpolate the feature map back to the same length of the input sequence. We see that in the first and second rows of Figure 6, the estimator trained on regular recording shows high saliency at the windows where cause and effect have interactions. The high saliency here indicates that the estimator captures highly relevant features to recognize a causal relationship in the system used to supervise, such as the time lag of cause-effect. In the third and fourth rows, we show the feature map of the estimator trained with noise-free data but received noisy input. Similar to the estimator in the ideal context, even if the input

data is mixed with noise, the feature map exhibits the estimator still pays special attention to the Windows cause-effect happens - as such the behavior of the estimator does make sense.

The precedence of cause is a core rule used to discriminate between cause and effect. Inspired by the time lag between cause and effect, we expect our estimator to give a different answer when it receives a modified input that violates the precedence of cause. Therefore, we move the effect transistor backward 30 time steps (one half-clock) and infer the relationship between this new pair. As shown in Figure 7, when the effect precedes the cause, the estimator rejects the previously recognized causal relationship, with very low confidence that it is causal. We suggest that our estimator understands the basic rule between cause and effect and has adopted it to do causal discovery since the rejection decision is consistent with the broken causal chain (see Appendix A.3 for more examples).

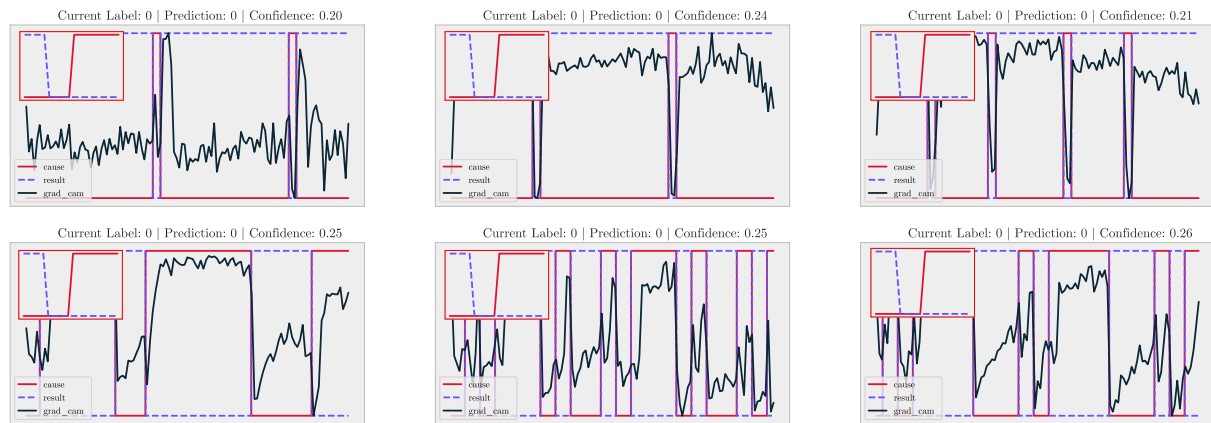

Figure 7: Cause-effect temporal reversal by making effect precedes cause.

## 4 Discussion

Here, we have demonstrated that domain-specific causal discovery from observational time-varying data can be framed as a machine learning problem. We use a causal system with numerous causal relations, the MOS 6502, for our main experiments. We find that the learned estimator significantly outperforms traditional human-designed methods, particularly in the additive noise domain. We also discover that learned causal discovery generalizes across different games. Lastly, we observe that the learned strategy, as expected, focuses on times when state transitions unfold.

Undoubtedly, the right causal discovery strategy will vary across domains. We test our methods on microprocessor causality, synthesized brain networks, and synthesized gene networks. We acknowledge that a method that works well on the microprocessor may not perform equally well in other real-world scenarios, such as in medicine or neuroscience. Nevertheless, we prototype our method on the microprocessor simply due to a lack of data in other domains: neither medicine nor neuroscience has large datasets of observational data along with ground truth perturbation-based causality data. As these datasets become available, revisiting learning approaches to causality will become crucial.

The right causal strategy also depends on different factors of data quality such as Signal to Noise Ratio (SNR). Real-world data contains noise, and often observations are only partial. Observations are also frequently short. The evolving internal state of other systems produces latent noise that will be difficult to learn. Interestingly, we find that adding large-scale observational noise, like 0.3 or 0.5, considerably degrades the estimator's performance, as the noise is so strong that it conceals or even reverses the original causal features, which is also common in machine learning (Dodge & Karam, 2016; Goodfellow et al., 2014) (see Appendix A.6). Moreover, we observe that in the small sample case, the Dream3 benchmark, the learning result is unsatisfactory. Developing causal discovery strategies that are large-scale noise tolerant and work within the constraints of relatively small datasets is an important issue for future research.

Our results are superior to those based on human assumptions (AUPRC $\sim 0.5$ vs humans at $\sim 0.2$). Furthermore, the results demonstrate a certain generalization ability even in a more limited observational setting and the potential for transferability in homogeneous systems. It is indeed a limitation that our learned estimator can only have a local view of a pair of elements in a system, which makes it challenging to acquire a broader perspective like other multivariate and graph-based causal discovery methods. However, due to the pairwise input form, our estimator can perform inference on arbitrary systems with different numbers of nodes, allowing it to conveniently transfer learned strategies to other systems and potentially to other domains. As such, it seems clear that the advantage of learning is significant in our system. Learning may thus make a considerable difference relative to the current state of the art in many fields like public policy or epidemiology. For instance, many perturbations in public policy, such as tax changes and exceptions from regulations, are performed every year. We can use the database containing all these perturbations and the resulting (noisy) ground truth measurements of perturbation effects in public policy for learning. Similarly, many perturbations in medicine occur every year, including the introduction of new drugs, new guidelines, and new procedures. We can also use the results from the thousands of such perturbation "experiments" to calibrate our approach.

Currently, algorithms for causal discovery from observational data are constructed using mathematical ideas and have strong theoretical support regarding identifiability. For example, the Hyvarinen approach is based on sparseness (Shimizu et al., 2006), and the Blei deconfounder is based on independent confounders (Wang & Blei, 2019). However, it is quite unclear how suitable these assumptions are and whether the causality they assume is appropriate for specific domains. Our approach can, in principle, discover ideas like deconfounders or sparse noise, but, importantly, it can use all of these ideas, those expressed by clever mathematicians and those that existed in specific domains but no one has yet discovered. Learning causal discovery promises to make the field more efficient.

The success of learning domain-specific causal discovery we have presented here suggests that we should use such approaches across various scientific disciplines. We advocate for large projects in the domains of public policy and epidemiology to produce datasets of ground-truth causality. A lack of proper benchmarks has long been holding back these fields. Our paper highlights an additional motivation for why we should produce such datasets: they promise to considerably improve the causal discovery procedures that power these fields.

**Broader Impact Statement**

Being able to efficiently do causal discovery, promises to make AI more useful to a larger group of scientists and engineers, which can deeply enrich domains like biology, economics, sociology, and even political science. Our research indicates a potential paradigm shift towards a data-driven approach to causal discovery, potentially revolutionizing our grasp of complex systems and phenomena. This interdisciplinary infusion promises to solve complex, long-standing problems by revealing not just correlations, but actual causative factors.

Along with this, there may be shifts in the balance between humans and machines, potentially with negative consequences. Mistakenly inferred causal relationships can have severe consequences, especially in critical fields like healthcare, finance, or environmental science. The misinterpretation of these inferences can lead to misguided policies, incorrect treatments, or flawed strategic decisions. Moreover, malicious actors, equipped with refined causal discovery tools, might exploit systems or manipulate outcomes more efficiently. For instance, understanding causal relationships could help in pinpointing vulnerabilities in financial and political systems, leading to more sophisticated attacks. The power to learn causal discovery with perturbation data also comes with ethical implications. For example, in social sciences or behavioral studies, perturbation experiments might lead to targeted interventions that could infringe on individual rights or privacy.

**Author Contributions**

Both authors conceptualized the ideas together and wrote the text. XW wrote the code and performed the analyses.

**Code Availability**

Code for simulation and learning is available at (https://github.com/KordingLab/LearningCausalDiscovery).

**Acknowledgments**

We thank Tony Liu for valuable discussions and comments on the project and writing. We thank Eric Jonas for sharing the source code for MOS 6502 simulation. We thank NIH grant 1-R01-EB028162-01 for support.

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

# A  Appendix

## A.1  Experiment details

### A.1.1  Architecture and training details

- Input: We set sequence length $L$ to be 3840 to align with the original 128 half-clock sequences. The window length is 32 with no overlapping and the dimensional of window embedding $C$ is 192.

- Encoder: We use a Transformer with depth 8, hidden size 768 and 3 attention heads, as sequence encoder. The attention pooler uses a two-layer MLP to calculate the attention score. We augment the input data by randomly shifting it in the range of [-1200, 1200].

- Optimization: Pooler output is regarded as $P(Causal|\boldsymbol{X})$ and compared with the adjacency matrix to acquire focal loss. We use focal loss (Lin et al., 2017) instead of regular binary cross-entropy loss with 0.7 as $\alpha$ and 3 as $\gamma$, because the sample is not balanced. We optimize the network with AdamW and a learning rate 0.001 and weight decay 0.05 and batch size 1024 for 100 epochs. The learning rate is adjusted by the Cosine Annealing scheduler in the rest process.

### A.1.2  MOS6502 settings

In the MOS 6502 experiments, we use relatively fast pair-wise methods including Pearson Correlation (Corr), Mutual Information (MI), VAR-LiNGAM (LiNGAM), and Linear Granger Causality (GC) to compare with our method. Correlation is a statistical measure that indicates the extent to which two variables change in relation to each other. Mutual Information is a metric quantifying the amount of information obtained about one variable through observing the other variable. Linear Granger Causality is a method to determine whether past values of one variable can linearly predict future values of another variable, indicating a causal relationship. VAR-LiNGAM integrates Vector Autoregression (VAR) with LiNGAM (Linear Non-Gaussian Acyclic Model) to identify causal relationships among multiple time series variables while considering non-Gaussian data distributions. Correlation and Mutual Information are often used to measure connectivity and causality in some fields like Neuroscience. Linear Granger Causality and VAR-LiNGAM are classical causal discovery algorithms for time series. Due to the potentially large graph size (hundreds of nodes), the graph-based methods are very time-consuming. AUROC, or Area Under the Receiver Operating Characteristic curve, is the area under the curve that plots the true positive rate (sensitivity) against the false positive rate (1 - specificity) at various threshold settings. It is a comprehensive indicator of a model's performance in a binary classification problem. AUPRC, or Area Under the Precision-Recall Curve, is the area under the curve that plots Precision (proportion of true positives over all positive predictions) against Recall (proportion of true positives over all actual positives) at various threshold settings. This metric is especially useful in binary classification problems with imbalanced classes

We calculate AUROC and AUPRC on all the methods mentioned above. For correlation, we calculate the AUROC and AUPRC on the absolute value of the Pearson correlation (implementation from Sklearn (Buitinck et al., 2013)). For the Mutual Information, we take the implementation from Wu et al. (2020) which optimizes for float value input, and calculate AUROC and AUPRC on it. For the VAR-LiNGAM, we take the implementations from LiNGAM (https://lingam.readthedocs.io/en/latest/tutorial/var.html) and use default settings except 5 as maximum time lag. For the Linear Granger Causality, we define the causal strength using the ratio of variances of the error from linear regression on bi-variate and auto-regression, and calculate AUROC and AUPRC on it. For our procedure, we calculate the AUROC directly on the network's output probability. All the operations mentioned above are done in a pair-wise way.

### A.1.3  NetSim settings

In the Netsim experiments, each simulation represents a kind of temporal property and has a different number of nodes. Each simulation has a number of subjects (trials) that share the same causal graph. Here we break the graph of each subject into pairs to get samples for pair-wise methods and directly use the graph of each subject as the input of graph-based methods. AUROC and AUPRC are both computed first on each subject and then averaged across subjects on the same simulation.

We use Correlation (Corr), DYNOTEARS (DYNO) (Pamfil et al., 2020), Multivariate Granger Causality (GC) Shojaie & Michailidis (2010), Mutual Information (MI), PCMCI+ (Peters et al., 2013), VAR-LiNGAM (LiNGAM) (Hyvärinen et al., 2010), Neural Granger Causality (cMLP, cLSTM) (Tank et al., 2021), eSRU and SRU (Khanna & Tan, 2019) to compare with our method. For Correlation, Mutual Information and VAR-LiNGAM, we use the same implementations as the experiments in the MOS 6502. For the Multivariate Granger Causality, we take the implementation from causal-learn (https://github.com/py-why/causal-learn) and use 2 as the time lag. For Granger Causality and VAR-LiNGAM, we merge the graphs from different time lag by using the entry with the largest absolute weight value. For the DYNOTEARS, we use the implementation from CausalNex (https://github.com/quantumblacklabs/causalnex) and parameter setting from Gong et al. (2022). For the PCMCI+, we use the implementation from tigramite (https://github.com/jakobrunge/tigramite) and the parameter setting from Gong et al. (2022). We merge the graphs from different time lags by using the maximum entry across time lags (it would be causal if any time lag present causality). For the Neural Granger Causality, eSRU, and SRU, we take the implementation and hyperparameter recommendation from (Tank et al., 2021) and Khanna & Tan (2019). With reference to how Löwe et al. (2022) compute AUROC, we first divide ten equally spaced values within the regularization parameter range from Khanna & Tan (2019). Then calculate the score of each edge by choosing the minimum regularization parameter value that makes the edge not existed as the edge score. And use the score matrix to compute AUROC and AUPRC, where the larger the edge score is, the higher probability the edge is causal. For our method, we use 16 as the window length and 128 as the window embedding dimension. We set the hidden size to be 512 and 4 attention heads. We optimize the network with a learning rate of 0.0005 with a batch size of 256. We augment the data by random shifting in a range of $[-100, 100]$. Other parameter settings are the same as Appendix A.1.1. All the diagonal entries are ignored (masked) when calculating AUROC to not consider self-loop.

### A.1.4 Dream3 settings

In the Dream3 experiments, the setting is quite different from MOS 6502 and NetSim. Dream3 consists of 5 gene networks and their corresponding recordings (Prill et al., 2010). The recording consists of 46 short trajectories, each length is 21. We take the same implementations and parameter settings from the same place as NetSim settings. We follow Khanna & Tan (2019) to center and upscale the raw data. For Correlation, Mutual Information, DYNOTEARS, PCMCI+ and Multivariate Granger Causality, we follow the reference to use each trajectory as an example and then merge the inferred graphs from all 46 trajectories into one graph by using the largest graph entry across trajectories (If there is an edge at any trajectory, then the edge exists). For Neural Granger Causality (cMLP and cLSTM) and eSRU and SRU, we stack 46 trajectories into a batch of data since they share the same causal dynamics. And take the norm of the network weight matrix as the inferred causal graph, which is the same as what Multivariate Granger Causality does. For our method, we directly train it and infer on the recording with all 46 trajectories and use 21 as the window size. So the output is directly the inference result of the corresponding recording. For our method, we use 21 as the window length and 64 as the window embedding dimension. We set the hidden size to be 64 and 4 attention heads. We optimize the network with a learning rate of 0.0005 with a batch size of 128. We augment the data by random shifting in a range of $[-322, 322]$. Other parameter settings are the same as Appendix A.1.1. All the diagonal entries are ignored (masked) when calculating AUROC to not consider self-loop.

### A.1.5 Grad-CAM settings

In the explanation study, we modified the implementations of Grad-CAM from Gildenblat & contributors (2021) to align with the temporal sequences instead of images. All gradients are min-max normalized to the range from 0 to 1. The showed examples in the main text and appendix are all taken from the period $[640, 768]$.

## A.2 Distribution of the causal effects in the MOS 6502

Here we show some data analysis of MOS 6502 while we are doing simulations and perturbations. It can be seen from the first subplot that there are many redundant transistors in the microprocessor but the number

of them varies across time. The number of unique transistors centers at around 400 600 in the most of time. In the second subplot, we show that the histogram plot of the number of positive pairs across time. It indicates that the number of positive pairs changes across time but centers at around 200 most of the time. Compared to the number of total pairs ($400^2$-$600^2$), it also shows the sparsity of causality in the large system. The third subplot shows how the causal graph (positive pairs number) changes as the microprocessor is running. The red dot is the minor case that positive samples are more than 400, which corresponds to what we see in the positive pairs histogram plot. Interestingly, we find that the presence of more than 400 positive samples is like a periodic activity with every three periods (384 half-clocks) as the interval. Notice that there is a TIA chip connecting and interacting with the microprocessor in a fixed frequency but gets ignored in this work, we think the TIA chip here is a hidden confounder to the whole microprocessor system and change the causal graph of the microprocessor as the interactions happen.

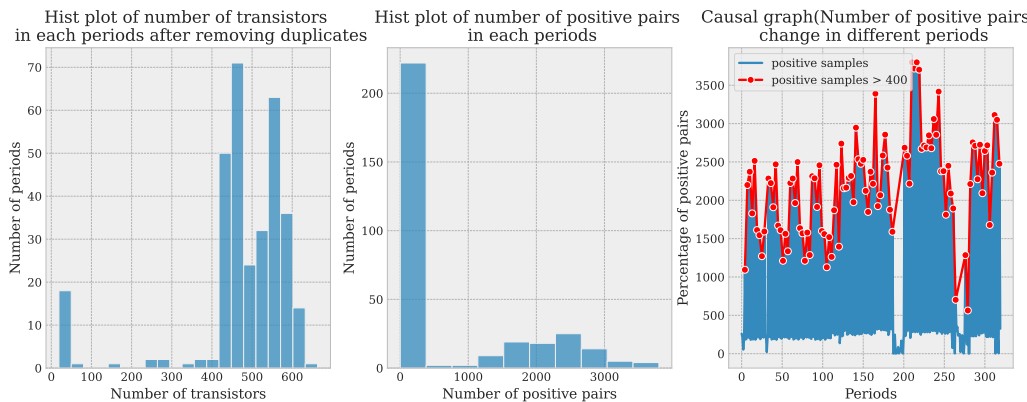

Figure 8: Distribution of the positive samples (causal graph) across time

### A.3 Supplementary materials of Grad-CAM and temporal reversal

Here we give more examples on the Grad-CAM result and temporal reversal. Figure 9 shows the gradient saliency plotting on the test set of period $[640, 768]$. It is clear that the gradient is more centered and active on the place where the transition between transistors happens, whether under noise or not.

### A.4 Supplementary materials of NetSim result

In the main text, we are more interested in AUPRC thus moving the AUROC comparison to here. Table 13 shows the AUROC comparison on the selected simulations without additive observational noise, which corresponds to the AUPRC comparison in Table 3. Table 4 shows the AUROC comparison on the selected simulations under additive observational noise with a scale 0.1, which corresponds to the AUPRC comparison in Table 4.

Figure 11 and Figure 12 show a part of our estimator's inferred result and the corresponding ground truth adjacency matrix. To help visualization, we use zero to fill up the diagonal entries of both the ground truth and inferred adjacency matrix. Entries of inferred results are the probability of causal relationships given pair-wise input, while the ground truth adjacency matrix is a binary matrix.

### A.5 Supplementary materials of Dream3 result

In the main text, we are more interested in AUPRC thus move the AUROC comparison to here. Table 8 shows the AUROC comparison on the selected datasets, which corresponds to the AUPRC comparison in Table 5.

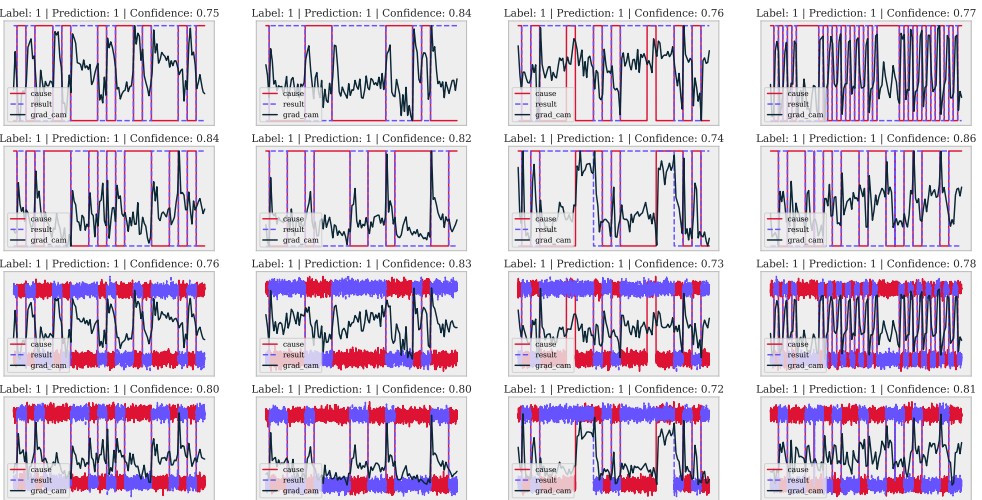

Figure 9: Grad-CAM Supplementary result in MOS 6502 for some other samples

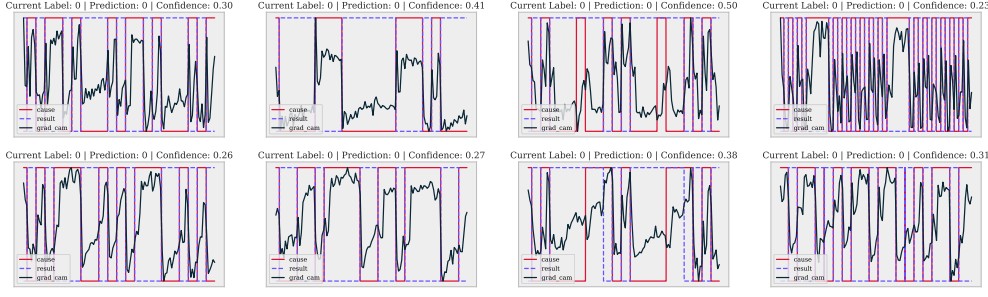

Figure 10: Temporal reversal in MOS 6502 for some other samples

Table 6: AUROC Comparison on different simulations of NetSim (mean ± std).

| Dataset | Corr | DYNO | GC | MI | PCMCI+ | LiNGAM | cMLP | cLSTM | eSRU | SRU | Transformer |
|---|---|---|---|---|---|---|---|---|---|---|---|
| Sim1 | $0.79 \pm 0.04$ | $0.73 \pm 0.08$ | $0.67 \pm 0.11$ | $0.66 \pm 0.11$ | $0.64 \pm 0.12$ | $0.66 \pm 0.13$ | $0.65 \pm 0.12$ | $0.64 \pm 0.12$ | $0.63 \pm 0.13$ | $0.62 \pm 0.13$ | $\mathbf{0.99 \pm 0.02}$ |
| Sim2 | $0.88 \pm 0.04$ | $0.81 \pm 0.08$ | $0.75 \pm 0.11$ | $0.73 \pm 0.11$ | $0.70 \pm 0.12$ | $0.69 \pm 0.12$ | $0.68 \pm 0.11$ | $0.67 \pm 0.11$ | $0.66 \pm 0.12$ | $0.66 \pm 0.11$ | $\mathbf{0.99 \pm 0.01}$ |
| Sim3 | $0.91 \pm 0.03$ | $0.85 \pm 0.07$ | $0.77 \pm 0.14$ | $0.76 \pm 0.13$ | $0.73 \pm 0.13$ | $0.73 \pm 0.12$ | $0.72 \pm 0.12$ | $0.70 \pm 0.12$ | $0.69 \pm 0.12$ | $0.68 \pm 0.12$ | $\mathbf{0.98 \pm 0.01}$ |
| Sim8 | $0.69 \pm 0.10$ | $0.66 \pm 0.10$ | $0.63 \pm 0.11$ | $0.62 \pm 0.11$ | $0.61 \pm 0.11$ | $0.62 \pm 0.12$ | $0.62 \pm 0.12$ | $0.61 \pm 0.12$ | $0.61 \pm 0.13$ | $0.61 \pm 0.13$ | $\mathbf{0.94 \pm 0.05}$ |
| Sim10 | $0.81 \pm 0.03$ | $0.69 \pm 0.12$ | $0.65 \pm 0.15$ | $0.67 \pm 0.14$ | $0.66 \pm 0.15$ | $0.66 \pm 0.16$ | $0.65 \pm 0.16$ | $0.64 \pm 0.16$ | $0.64 \pm 0.16$ | $0.63 \pm 0.16$ | $\mathbf{0.98 \pm 0.02}$ |
| Sim11 | $0.78 \pm 0.03$ | $0.77 \pm 0.04$ | $0.72 \pm 0.09$ | $0.70 \pm 0.08$ | $0.68 \pm 0.10$ | $0.68 \pm 0.10$ | $0.67 \pm 0.09$ | $0.66 \pm 0.10$ | $0.65 \pm 0.10$ | $0.65 \pm 0.10$ | $\mathbf{0.93 \pm 0.03}$ |
| Sim12 | $0.87 \pm 0.03$ | $0.83 \pm 0.05$ | $0.76 \pm 0.12$ | $0.73 \pm 0.12$ | $0.70 \pm 0.13$ | $0.69 \pm 0.13$ | $0.68 \pm 0.12$ | $0.66 \pm 0.13$ | $0.66 \pm 0.13$ | $0.65 \pm 0.13$ | $\mathbf{0.98 \pm 0.02}$ |
| Sim13 | $0.66 \pm 0.10$ | $0.66 \pm 0.08$ | $0.62 \pm 0.10$ | $0.62 \pm 0.12$ | $0.59 \pm 0.12$ | $0.59 \pm 0.12$ | $0.59 \pm 0.12$ | $0.59 \pm 0.12$ | $0.59 \pm 0.12$ | $0.58 \pm 0.12$ | $\mathbf{0.72 \pm 0.11}$ |
| Sim14 | $0.80 \pm 0.03$ | $0.74 \pm 0.08$ | $0.69 \pm 0.10$ | $0.67 \pm 0.10$ | $0.64 \pm 0.11$ | $0.66 \pm 0.13$ | $0.65 \pm 0.13$ | $0.64 \pm 0.13$ | $0.63 \pm 0.14$ | $0.62 \pm 0.15$ | $\mathbf{0.97 \pm 0.03}$ |
| Sim15 | $0.74 \pm 0.05$ | $0.68 \pm 0.07$ | $0.64 \pm 0.12$ | $0.66 \pm 0.12$ | $0.66 \pm 0.12$ | $0.70 \pm 0.16$ | $0.68 \pm 0.16$ | $0.67 \pm 0.16$ | $0.65 \pm 0.16$ | $0.64 \pm 0.16$ | $\mathbf{0.88 \pm 0.05}$ |
| Sim16 | $0.70 \pm 0.03$ | $0.64 \pm 0.07$ | $0.60 \pm 0.10$ | $0.60 \pm 0.09$ | $0.59 \pm 0.09$ | $0.60 \pm 0.11$ | $0.59 \pm 0.11$ | $0.59 \pm 0.11$ | $0.58 \pm 0.12$ | $0.58 \pm 0.12$ | $\mathbf{0.97 \pm 0.02}$ |
| Sim17 | $0.91 \pm 0.02$ | $0.87 \pm 0.05$ | $0.78 \pm 0.14$ | $0.78 \pm 0.13$ | $0.76 \pm 0.13$ | $0.78 \pm 0.13$ | $0.77 \pm 0.13$ | $0.75 \pm 0.14$ | $0.73 \pm 0.14$ | $0.71 \pm 0.15$ | $\mathbf{1.00 \pm 0.00}$ |
| Sim18 | $0.80 \pm 0.04$ | $0.74 \pm 0.08$ | $0.68 \pm 0.15$ | $0.67 \pm 0.14$ | $0.64 \pm 0.14$ | $0.67 \pm 0.15$ | $0.65 \pm 0.16$ | $0.64 \pm 0.16$ | $0.63 \pm 0.16$ | $0.62 \pm 0.16$ | $\mathbf{0.99 \pm 0.01}$ |
| Sim21 | $0.80 \pm 0.05$ | $0.74 \pm 0.08$ | $0.68 \pm 0.12$ | $0.65 \pm 0.12$ | $0.63 \pm 0.12$ | $0.65 \pm 0.13$ | $0.64 \pm 0.13$ | $0.62 \pm 0.13$ | $0.62 \pm 0.13$ | $0.61 \pm 0.13$ | $\mathbf{0.98 \pm 0.03}$ |
| Sim22 | $\mathbf{0.69 \pm 0.06}$ | $0.66 \pm 0.07$ | $0.62 \pm 0.13$ | $0.63 \pm 0.12$ | $0.61 \pm 0.12$ | $0.60 \pm 0.12$ | $0.58 \pm 0.13$ | $0.57 \pm 0.13$ | $0.56 \pm 0.13$ | $0.56 \pm 0.13$ | $0.39 \pm 0.14$ |
| Sim23 | $0.67 \pm 0.07$ | $0.64 \pm 0.06$ | $0.62 \pm 0.09$ | $0.63 \pm 0.10$ | $0.65 \pm 0.11$ | $0.68 \pm 0.15$ | $0.67 \pm 0.15$ | $0.65 \pm 0.15$ | $0.63 \pm 0.16$ | $0.61 \pm 0.17$ | $\mathbf{0.72 \pm 0.07}$ |
| Sim24 | $\mathbf{0.58 \pm 0.09}$ | $0.53 \pm 0.11$ | $0.54 \pm 0.11$ | $0.56 \pm 0.11$ | $0.57 \pm 0.12$ | $0.57 \pm 0.12$ | $0.55 \pm 0.13$ | $0.55 \pm 0.13$ | $0.55 \pm 0.13$ | $0.54 \pm 0.13$ | $0.56 \pm 0.12$ |

Table 7: AUROC comparison on NetSim under noise with scale 0.1 (mean ± std).

| Dataset | Corr | DYNO | GC | MI | PCMCI+ | LiNGAM | cMLP | cLSTM | eSRU | SRU | Transformer |
|---------|------|------|-----|-----|--------|--------|------|-------|------|-----|-------------|
| Sim1 | 0.72 ± 0.07 | 0.66 ± 0.09 | 0.64 ± 0.12 | 0.63 ± 0.11 | 0.61 ± 0.11 | 0.62 ± 0.13 | 0.61 ± 0.13 | 0.61 ± 0.14 | 0.60 ± 0.14 | 0.59 ± 0.14 | **0.87±0.13** |
| Sim2 | 0.78 ± 0.04 | 0.70 ± 0.09 | 0.66 ± 0.11 | 0.65 ± 0.11 | 0.62 ± 0.11 | 0.64 ± 0.11 | 0.63 ± 0.11 | 0.62 ± 0.11 | 0.62 ± 0.11 | 0.61 ± 0.11 | **0.90±0.05** |
| Sim3 | 0.82 ± 0.08 | 0.72 ± 0.11 | 0.68 ± 0.12 | 0.65 ± 0.12 | 0.62 ± 0.12 | 0.63 ± 0.11 | 0.63 ± 0.10 | 0.62 ± 0.10 | 0.62 ± 0.10 | 0.61 ± 0.09 | **0.89±0.06** |
| Sim8 | 0.65 ± 0.07 | 0.61 ± 0.08 | 0.62 ± 0.10 | 0.60 ± 0.11 | 0.58 ± 0.11 | 0.59 ± 0.11 | 0.60 ± 0.12 | 0.60 ± 0.12 | 0.60 ± 0.12 | 0.59 ± 0.12 | **0.82±0.13** |
| Sim10 | 0.75 ± 0.07 | 0.64 ± 0.12 | 0.64 ± 0.13 | 0.63 ± 0.13 | 0.61 ± 0.13 | 0.62 ± 0.13 | 0.62 ± 0.13 | 0.62 ± 0.14 | 0.61 ± 0.14 | 0.60 ± 0.15 | **0.84±0.13** |
| Sim11 | 0.71 ± 0.04 | 0.69 ± 0.06 | 0.66 ± 0.08 | 0.63 ± 0.10 | 0.61 ± 0.10 | 0.62 ± 0.09 | 0.62 ± 0.09 | 0.61 ± 0.09 | 0.61 ± 0.09 | 0.60 ± 0.09 | **0.82±0.06** |
| Sim12 | 0.76 ± 0.04 | 0.71 ± 0.07 | 0.67 ± 0.09 | 0.64 ± 0.10 | 0.61 ± 0.11 | 0.62 ± 0.11 | 0.62 ± 0.10 | 0.61 ± 0.10 | 0.60 ± 0.11 | 0.59 ± 0.11 | **0.88±0.06** |
| Sim13 | 0.58 ± 0.09 | 0.57 ± 0.09 | 0.57 ± 0.11 | 0.55 ± 0.13 | 0.54 ± 0.12 | 0.56 ± 0.14 | 0.56 ± 0.13 | 0.57 ± 0.13 | 0.56 ± 0.14 | 0.56 ± 0.13 | **0.58±0.14** |
| Sim14 | 0.75 ± 0.07 | 0.67 ± 0.10 | 0.65 ± 0.11 | 0.61 ± 0.13 | 0.59 ± 0.13 | 0.60 ± 0.13 | 0.60 ± 0.13 | 0.60 ± 0.13 | 0.59 ± 0.13 | 0.58 ± 0.13 | **0.90±0.09** |
| Sim15 | 0.67 ± 0.06 | 0.61 ± 0.08 | 0.63 ± 0.11 | 0.63 ± 0.10 | 0.62 ± 0.10 | 0.64 ± 0.11 | 0.64 ± 0.11 | 0.63 ± 0.11 | 0.62 ± 0.12 | 0.61 ± 0.12 | **0.84±0.10** |
| Sim16 | 0.65 ± 0.05 | 0.59 ± 0.08 | 0.58 ± 0.11 | 0.57 ± 0.11 | 0.56 ± 0.11 | 0.56 ± 0.12 | 0.56 ± 0.12 | 0.56 ± 0.12 | 0.56 ± 0.12 | 0.56 ± 0.12 | **0.86±0.07** |
| Sim17 | 0.85 ± 0.03 | 0.77 ± 0.09 | 0.73 ± 0.11 | 0.69 ± 0.12 | 0.67 ± 0.12 | 0.68 ± 0.12 | 0.67 ± 0.11 | 0.66 ± 0.11 | 0.65 ± 0.11 | 0.65 ± 0.11 | **0.96±0.03** |
| Sim18 | 0.75 ± 0.06 | 0.67 ± 0.10 | 0.65 ± 0.10 | 0.61 ± 0.13 | 0.59 ± 0.12 | 0.61 ± 0.13 | 0.60 ± 0.13 | 0.60 ± 0.14 | 0.59 ± 0.14 | 0.59 ± 0.14 | **0.89±0.12** |
| Sim21 | 0.71 ± 0.08 | 0.67 ± 0.08 | 0.65 ± 0.10 | 0.63 ± 0.11 | 0.61 ± 0.11 | 0.61 ± 0.13 | 0.60 ± 0.13 | 0.60 ± 0.13 | 0.60 ± 0.13 | 0.60 ± 0.14 | **0.86±0.09** |
| Sim22 | **0.65±0.07** | 0.64 ± 0.08 | 0.62 ± 0.11 | 0.61 ± 0.11 | 0.59 ± 0.11 | 0.58 ± 0.12 | 0.58 ± 0.12 | 0.57 ± 0.12 | 0.57 ± 0.12 | 0.57 ± 0.12 | 0.50 ± 0.15 |
| Sim23 | 0.58 ± 0.07 | 0.54 ± 0.07 | 0.55 ± 0.10 | 0.55 ± 0.10 | 0.56 ± 0.12 | 0.57 ± 0.12 | 0.57 ± 0.12 | 0.57 ± 0.12 | 0.56 ± 0.12 | 0.55 ± 0.12 | **0.69±0.07** |
| Sim24 | 0.49 ± 0.04 | 0.49 ± 0.04 | 0.49 ± 0.10 | 0.49 ± 0.10 | 0.50 ± 0.10 | 0.49 ± 0.11 | 0.50 ± 0.11 | 0.50 ± 0.11 | **0.50±0.11** | 0.49 ± 0.11 | 0.49 ± 0.04 |

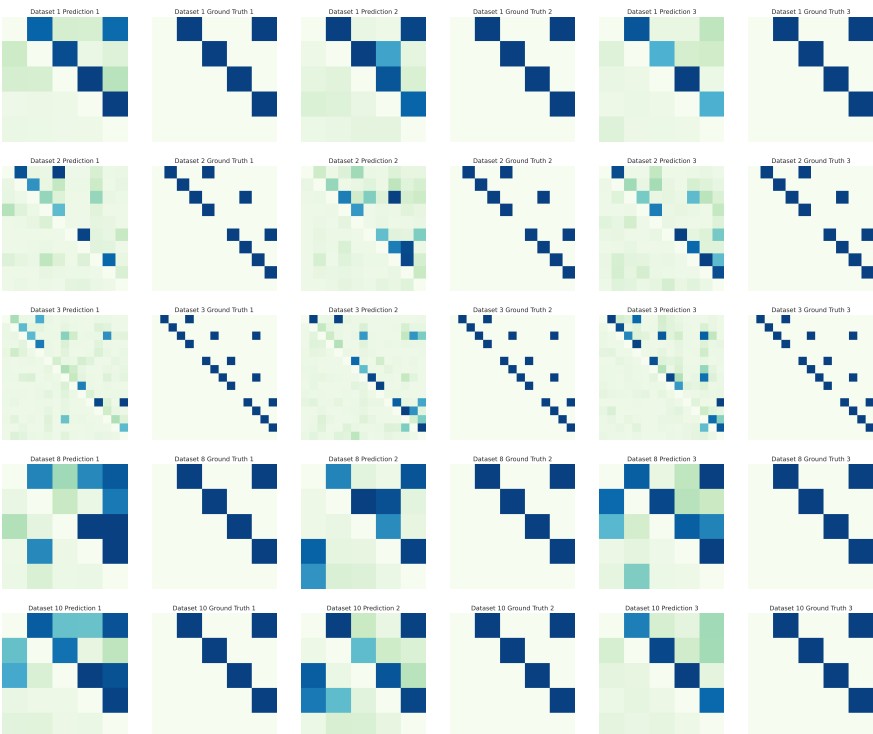

Figure 11: Ground truth adjacency matrix and inferred adjacency matrix examples on Simulation 1, 3, 8, 10 in NetSim (Row 0, 1, 2, 3 represent Simulation 1, 3, 8, 10 in order. Column 1, 3, 5 shows raw ground truth connection strength matrix, column 0, 2, 4 inferred matrix shows the inferred $P(Causal|\boldsymbol{X})$)

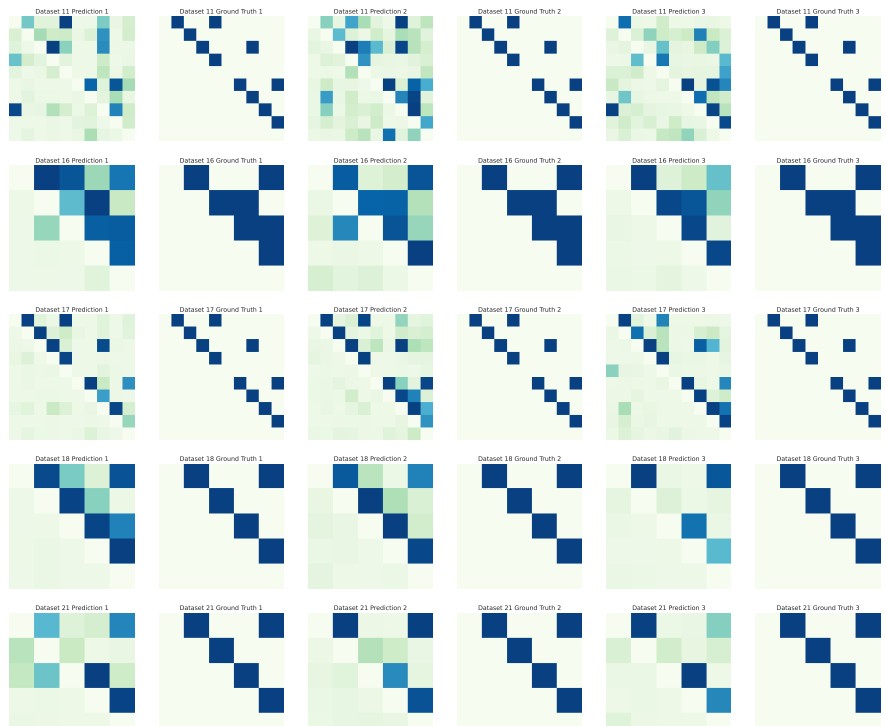

Figure 12: Ground truth adjacency matrix and inferred adjacency matrix examples on Simulation $11, 16, 17, 18, 21$ in NetSim (Row $0, 1, 2, 3$ represent Simulation $11, 16, 17, 18, 21$ in order. Column $1, 3, 5$ shows raw ground truth connection strength matrix, column $0, 2, 4$ inferred matrix shows the inferred $P(Causal|\boldsymbol{X})$

Table 8: AUROC comparison on the Dream3

| Dataset | Corr | DYNO | GC | MI | PCMCI+ | cMLP | cLSTM | eSRU | SRU | Transformer |
|---------|------|------|----|----|--------|------|-------|------|-----|-------------|
| Ecoli2 | 0.52 | 0.50 | 0.52 | 0.53 | 0.47 | 0.51 | 0.58 | 0.54 | 0.55 | **0.73** |
| Yeast2 | 0.49 | 0.50 | 0.53 | 0.52 | 0.50 | 0.46 | 0.49 | 0.49 | 0.49 | **0.58** |
| Yeast3 | 0.49 | 0.50 | 0.46 | 0.48 | 0.49 | 0.50 | **0.57** | 0.52 | 0.52 | 0.53 |

### A.6 Supplementary noise result on MOS 6502 and NetSim

The main text focuses on analyzing small-scale noise $(0.03, 0.05, 0.1)$ on MOS 6502 and NetSim and shows our learned estimator's robust noise-tolerate ability. However, it is known in the Machine Learning field that noise in the input data (e.g., images) can impact the performance of machine learning models (Dodge & Karam, 2016; Goodfellow et al., 2014). To further explore our estimator's limitation under observational noise, we evaluate our estimator under much stronger noise on MOS 6502 and NetSim. We adopt the same experiment settings as A.1.2 and A.1.3, except for using noise with scales of 0.3 and 0.5.

#### A.6.1 Extra noise result on MOS 6502

Here we show the evaluation result on MOS 6502 under noise of scale 0.3 and 0.5, specifically on Donkey Kong. We see that under a noise scale of 0.3, even though it still outperforms other methods in most cases, its ability to recognize positive class degenerates a lot. It becomes more severe when the noise scale is increased to 0.5, which is surpassed by other methods in four periods. We think that the noise is so large that it severely covers the original causal patterns learned by the estimator, which makes the estimator wrongly estimate the causal relationships. It is also the case that the transistor signal is sparse and there is no noise flowing inside the system, making it relatively less information to refer to when there exists a large-scale noise.

Table 9: Models' performance on Donkey Kong under noise of scale 0.3 and 0.5.

| Noise Scale | Period | AUROC | | | | | AUPRC | | | | |
|---|---|---|---|---|---|---|---|---|---|---|---|
| | | Corr | MI | LiNGAM | GC | Transformer | Corr | MI | LiNGAM | GC | Transformer |
| 0.3 | $[0, 128]$ | 0.95 | 0.94 | 0.62 | **0.96** | 0.93 | 0.11 | 0.13 | 0.00 | **0.14** | 0.08 |
| | $[128, 256]$ | 0.89 | 0.89 | 0.53 | 0.89 | **0.95** | 0.12 | **0.13** | 0.00 | 0.11 | 0.04 |
| | $[384, 512]$ | 0.88 | 0.89 | 0.47 | 0.89 | **0.94** | 0.07 | 0.07 | 0.00 | 0.09 | **0.10** |
| | $[512, 640]$ | 0.63 | 0.65 | 0.59 | 0.62 | **0.90** | 0.04 | 0.05 | 0.01 | 0.04 | **0.28** |
| | $[640, 768]$ | 0.94 | 0.94 | 0.70 | **0.94** | 0.90 | 0.17 | **0.17** | 0.01 | 0.13 | 0.08 |
| | $mean \pm std$ | $0.86 \pm 0.12$ | $0.86 \pm 0.11$ | $0.58 \pm 0.08$ | $0.86 \pm 0.12$ | **0.92±0.02** | $0.10 \pm 0.04$ | $0.11 \pm 0.04$ | $0.00 \pm 0.00$ | $0.10 \pm 0.04$ | **0.12±0.08** |
| 0.5 | $[0, 128]$ | 0.92 | 0.91 | 0.35 | **0.93** | 0.64 | 0.10 | 0.08 | 0.00 | **0.11** | 0.01 |
| | $[128, 256]$ | 0.87 | **0.89** | 0.32 | 0.88 | 0.68 | 0.13 | 0.12 | 0.00 | **0.14** | 0.01 |
| | $[384, 512]$ | 0.85 | 0.75 | 0.27 | **0.86** | 0.81 | 0.05 | **0.06** | 0.00 | 0.05 | 0.05 |
| | $[512, 640]$ | 0.63 | 0.62 | 0.51 | 0.64 | **0.78** | 0.03 | 0.03 | 0.01 | 0.03 | **0.15** |
| | $[640, 768]$ | 0.92 | 0.89 | 0.28 | **0.93** | 0.55 | **0.18** | 0.12 | 0.00 | 0.11 | 0.00 |
| | $mean \pm std$ | $0.84 \pm 0.11$ | $0.81 \pm 0.11$ | $0.35 \pm 0.09$ | **0.85±0.11** | $0.69 \pm 0.09$ | **0.10±0.05** | $0.08 \pm 0.03$ | $0.00 \pm 0.00$ | $0.09 \pm 0.04$ | $0.04 \pm 0.06$ |

#### A.6.2 Extra noise result on NetSim

Here we show the evaluation result on NetSim under noise of scale 0.3 and 0.5. Similar to MOS 6502, our learned estimator also degrades under large-scale noise. However, it still outperforms other baselines in most cases in noise scales 0.3 and 0.5. Especially when evaluated by AUPRC, our learned estimator surpass other methods in almost all cases. Compared to the large-scale noise experiment on MOS 6502, we see that it presents a better noise-tolerate ability. We think it is due to more dense causal effects that flow in the system, helping the estimator recognize causality under large-scale noise. Future work considers incorporating de-noise techniques and self-supervised learning to create a representation that is more robust to large-scale noise.

Table 10: AUROC Comparison on different simulations of NetSim under noise of scale 0.3 (mean ± std).

| Dataset | Corr | DYNO | GC | MI | PCMCI+ | LiNGAM | cMLP | cLSTM | eSRU | SRU | Transformer |
|---|---|---|---|---|---|---|---|---|---|---|---|
| Sim1 | 0.49 ± 0.05 | 0.50 ± 0.04 | **0.53±0.08** | 0.51 ± 0.10 | 0.51 ± 0.09 | 0.51 ± 0.10 | 0.52 ± 0.10 | 0.53 ± 0.10 | 0.51 ± 0.12 | 0.50 ± 0.12 | 0.50 ± 0.14 |
| Sim2 | 0.50 ± 0.03 | 0.49 ± 0.03 | 0.49 ± 0.03 | 0.51 ± 0.06 | 0.51 ± 0.05 | 0.51 ± 0.06 | 0.51 ± 0.05 | 0.51 ± 0.06 | 0.51 ± 0.06 | 0.50 ± 0.06 | **0.61±0.08** |
| Sim3 | 0.50 ± 0.02 | 0.50 ± 0.02 | 0.51 ± 0.03 | 0.52 ± 0.04 | 0.52 ± 0.04 | 0.53 ± 0.05 | 0.53 ± 0.05 | 0.53 ± 0.05 | 0.53 ± 0.05 | 0.53 ± 0.05 | **0.56±0.05** |
| Sim8 | 0.49 ± 0.06 | 0.50 ± 0.04 | 0.52 ± 0.09 | 0.52 ± 0.09 | 0.52 ± 0.08 | 0.52 ± 0.09 | 0.53 ± 0.10 | **0.54±0.10** | 0.54 ± 0.11 | 0.53 ± 0.11 | 0.49 ± 0.12 |
| Sim10 | 0.49 ± 0.07 | 0.49 ± 0.06 | 0.54 ± 0.10 | 0.53 ± 0.11 | 0.54 ± 0.10 | 0.54 ± 0.11 | 0.55 ± 0.11 | **0.56±0.11** | 0.56 ± 0.11 | 0.54 ± 0.11 | 0.52 ± 0.11 |
| Sim11 | 0.51 ± 0.04 | 0.51 ± 0.03 | 0.51 ± 0.03 | 0.52 ± 0.05 | 0.51 ± 0.05 | 0.51 ± 0.05 | 0.52 ± 0.05 | 0.51 ± 0.05 | 0.51 ± 0.05 | 0.51 ± 0.05 | **0.58±0.09** |
| Sim12 | 0.49 ± 0.03 | 0.49 ± 0.03 | 0.50 ± 0.03 | 0.49 ± 0.05 | 0.49 ± 0.05 | 0.50 ± 0.05 | 0.50 ± 0.06 | 0.49 ± 0.06 | 0.49 ± 0.06 | 0.49 ± 0.07 | **0.60±0.08** |
| Sim13 | 0.49 ± 0.04 | 0.50 ± 0.03 | **0.52±0.07** | 0.52 ± 0.10 | 0.52 ± 0.10 | 0.52 ± 0.11 | 0.52 ± 0.10 | 0.52 ± 0.10 | 0.52 ± 0.10 | 0.52 ± 0.10 | 0.38 ± 0.12 |
| Sim14 | **0.52±0.04** | 0.51 ± 0.04 | 0.49 ± 0.07 | 0.49 ± 0.08 | 0.49 ± 0.07 | 0.48 ± 0.09 | 0.47 ± 0.09 | 0.47 ± 0.09 | 0.47 ± 0.09 | 0.46 ± 0.09 | 0.51 ± 0.13 |
| Sim15 | 0.49 ± 0.05 | 0.49 ± 0.03 | 0.54 ± 0.09 | 0.52 ± 0.11 | 0.52 ± 0.10 | 0.53 ± 0.10 | 0.55 ± 0.10 | **0.56±0.10** | 0.55 ± 0.11 | 0.54 ± 0.11 | 0.52 ± 0.11 |
| Sim16 | 0.53 ± 0.05 | 0.51 ± 0.05 | 0.52 ± 0.09 | 0.53 ± 0.09 | 0.52 ± 0.08 | 0.52 ± 0.09 | 0.53 ± 0.09 | 0.54 ± 0.09 | 0.53 ± 0.10 | 0.52 ± 0.10 | **0.56±0.10** |
| Sim17 | 0.54 ± 0.06 | 0.52 ± 0.05 | 0.51 ± 0.05 | 0.52 ± 0.06 | 0.51 ± 0.05 | 0.51 ± 0.06 | 0.51 ± 0.06 | 0.51 ± 0.06 | 0.51 ± 0.06 | 0.51 ± 0.07 | **0.68±0.07** |
| Sim18 | 0.50 ± 0.04 | 0.51 ± 0.03 | **0.52±0.08** | 0.49 ± 0.10 | 0.50 ± 0.09 | 0.49 ± 0.10 | 0.49 ± 0.11 | 0.49 ± 0.11 | 0.49 ± 0.11 | 0.49 ± 0.11 | 0.50 ± 0.11 |
| Sim21 | 0.47 ± 0.05 | 0.49 ± 0.04 | 0.52 ± 0.08 | 0.53 ± 0.09 | 0.53 ± 0.08 | 0.51 ± 0.10 | 0.52 ± 0.10 | 0.52 ± 0.11 | 0.51 ± 0.11 | 0.50 ± 0.11 | **0.55±0.12** |
| Sim22 | 0.50 ± 0.05 | 0.49 ± 0.05 | 0.50 ± 0.08 | 0.50 ± 0.09 | 0.50 ± 0.08 | 0.48 ± 0.10 | 0.48 ± 0.10 | 0.48 ± 0.10 | 0.48 ± 0.10 | 0.47 ± 0.11 | **0.54±0.11** |
| Sim23 | 0.43 ± 0.04 | 0.47 ± 0.04 | 0.49 ± 0.09 | 0.50 ± 0.10 | 0.50 ± 0.09 | 0.49 ± 0.11 | 0.50 ± 0.11 | 0.50 ± 0.11 | 0.49 ± 0.12 | 0.48 ± 0.12 | **0.58±0.05** |
| Sim24 | 0.42 ± 0.05 | 0.46 ± 0.05 | 0.49 ± 0.08 | 0.47 ± 0.11 | 0.48 ± 0.10 | 0.48 ± 0.10 | 0.49 ± 0.11 | **0.51±0.11** | 0.49 ± 0.12 | 0.49 ± 0.12 | 0.46 ± 0.10 |

Table 11: AUPRC Comparison on different simulations of NetSim under noise of scale 0.3 (mean ± std).

| Dataset | Corr | DYNO | GC | MI | PCMCI+ | LiNGAM | cMLP | cLSTM | eSRU | SRU | Transformer |
|---|---|---|---|---|---|---|---|---|---|---|---|
| Sim1 | 0.26 ± 0.02 | 0.26 ± 0.02 | 0.32 ± 0.11 | 0.30 ± 0.10 | 0.30 ± 0.10 | 0.30 ± 0.09 | 0.31 ± 0.09 | 0.31 ± 0.09 | 0.30 ± 0.09 | 0.30 ± 0.09 | **0.43±0.13** |
| Sim2 | 0.13 ± 0.01 | 0.13 ± 0.01 | 0.13 ± 0.02 | 0.14 ± 0.04 | 0.14 ± 0.04 | 0.15 ± 0.04 | 0.15 ± 0.04 | 0.14 ± 0.04 | 0.14 ± 0.04 | 0.14 ± 0.04 | **0.21±0.08** |
| Sim3 | 0.09 ± 0.01 | 0.09 ± 0.01 | 0.10 ± 0.03 | 0.11 ± 0.03 | 0.11 ± 0.03 | 0.11 ± 0.03 | 0.11 ± 0.03 | 0.11 ± 0.03 | 0.11 ± 0.03 | 0.11 ± 0.03 | **0.14±0.02** |
| Sim8 | 0.27 ± 0.04 | 0.26 ± 0.03 | 0.31 ± 0.10 | 0.31 ± 0.09 | 0.31 ± 0.08 | 0.32 ± 0.09 | 0.32 ± 0.09 | 0.32 ± 0.09 | 0.32 ± 0.09 | 0.31 ± 0.09 | **0.38±0.10** |
| Sim10 | 0.28 ± 0.04 | 0.26 ± 0.04 | 0.32 ± 0.12 | 0.32 ± 0.11 | 0.33 ± 0.10 | 0.33 ± 0.10 | 0.34 ± 0.10 | 0.34 ± 0.09 | 0.33 ± 0.09 | 0.32 ± 0.09 | **0.39±0.11** |
| Sim11 | 0.14 ± 0.02 | 0.13 ± 0.02 | 0.14 ± 0.02 | 0.14 ± 0.03 | 0.14 ± 0.03 | 0.15 ± 0.04 | 0.15 ± 0.04 | 0.15 ± 0.04 | 0.14 ± 0.03 | 0.14 ± 0.03 | **0.18±0.05** |
| Sim12 | 0.13 ± 0.01 | 0.13 ± 0.01 | 0.13 ± 0.03 | 0.14 ± 0.03 | 0.13 ± 0.03 | 0.14 ± 0.04 | 0.14 ± 0.03 | 0.14 ± 0.03 | 0.14 ± 0.03 | 0.14 ± 0.03 | **0.21±0.08** |
| Sim13 | 0.36 ± 0.05 | 0.36 ± 0.05 | 0.41 ± 0.09 | 0.42 ± 0.11 | 0.41 ± 0.10 | **0.42±0.10** | 0.42 ± 0.10 | 0.42 ± 0.09 | 0.41 ± 0.09 | 0.41 ± 0.09 | 0.40 ± 0.08 |
| Sim14 | 0.27 ± 0.03 | 0.26 ± 0.03 | 0.27 ± 0.04 | 0.28 ± 0.05 | 0.27 ± 0.04 | 0.28 ± 0.05 | 0.27 ± 0.05 | 0.27 ± 0.05 | 0.27 ± 0.05 | 0.27 ± 0.05 | **0.36±0.10** |
| Sim15 | 0.27 ± 0.03 | 0.26 ± 0.02 | 0.33 ± 0.12 | 0.32 ± 0.11 | 0.31 ± 0.10 | 0.32 ± 0.10 | 0.32 ± 0.10 | 0.33 ± 0.09 | 0.32 ± 0.09 | 0.31 ± 0.09 | **0.41±0.12** |
| Sim16 | 0.38 ± 0.02 | 0.37 ± 0.02 | 0.41 ± 0.09 | 0.41 ± 0.08 | 0.40 ± 0.08 | 0.40 ± 0.08 | 0.41 ± 0.08 | 0.42 ± 0.09 | 0.41 ± 0.09 | 0.41 ± 0.09 | **0.46±0.09** |
| Sim17 | 0.15 ± 0.04 | 0.14 ± 0.03 | 0.15 ± 0.04 | 0.15 ± 0.04 | 0.14 ± 0.03 | 0.15 ± 0.04 | 0.15 ± 0.04 | 0.15 ± 0.04 | 0.15 ± 0.04 | 0.14 ± 0.04 | **0.26±0.08** |
| Sim18 | 0.27 ± 0.02 | 0.26 ± 0.02 | 0.31 ± 0.09 | 0.29 ± 0.09 | 0.29 ± 0.08 | 0.29 ± 0.09 | 0.30 ± 0.08 | 0.29 ± 0.08 | 0.29 ± 0.08 | 0.29 ± 0.08 | **0.41±0.12** |
| Sim21 | 0.26 ± 0.02 | 0.26 ± 0.01 | 0.32 ± 0.10 | 0.31 ± 0.10 | 0.31 ± 0.09 | 0.30 ± 0.09 | 0.31 ± 0.09 | 0.31 ± 0.09 | 0.30 ± 0.09 | 0.30 ± 0.09 | **0.45±0.11** |
| Sim22 | 0.27 ± 0.02 | 0.25 ± 0.02 | 0.29 ± 0.08 | 0.29 ± 0.08 | 0.28 ± 0.07 | 0.28 ± 0.07 | 0.28 ± 0.06 | 0.28 ± 0.07 | 0.28 ± 0.07 | 0.28 ± 0.07 | **0.41±0.14** |
| Sim23 | 0.25 ± 0.03 | 0.25 ± 0.02 | 0.29 ± 0.10 | 0.30 ± 0.09 | 0.30 ± 0.09 | 0.30 ± 0.09 | 0.30 ± 0.09 | 0.30 ± 0.09 | 0.29 ± 0.08 | 0.29 ± 0.08 | **0.42±0.10** |
| Sim24 | 0.24 ± 0.02 | 0.25 ± 0.02 | 0.28 ± 0.06 | 0.27 ± 0.06 | 0.27 ± 0.06 | 0.27 ± 0.06 | 0.28 ± 0.07 | 0.29 ± 0.07 | 0.29 ± 0.07 | 0.28 ± 0.07 | **0.33±0.07** |

Table 12: AUROC Comparison on different simulations of NetSim under noise of scale 0.5 (mean ± std).

| Dataset | Corr | DYNO | GC | MI | PCMCI+ | LiNGAM | cMLP | cLSTM | eSRU | SRU | Transformer |
|---|---|---|---|---|---|---|---|---|---|---|---|
| Sim1 | 0.48 ± 0.03 | 0.49 ± 0.02 | 0.50 ± 0.04 | 0.48 ± 0.07 | **0.50±0.08** | 0.47 ± 0.11 | 0.48 ± 0.10 | 0.48 ± 0.10 | 0.47 ± 0.11 | 0.47 ± 0.11 | 0.48 ± 0.13 |
| Sim2 | 0.48 ± 0.01 | 0.49 ± 0.01 | 0.50 ± 0.03 | 0.49 ± 0.05 | 0.49 ± 0.04 | 0.49 ± 0.04 | 0.49 ± 0.04 | 0.50 ± 0.04 | 0.49 ± 0.04 | 0.49 ± 0.04 | **0.51±0.07** |
| Sim3 | 0.47 ± 0.01 | 0.49 ± 0.02 | 0.51 ± 0.05 | 0.51 ± 0.04 | 0.51 ± 0.04 | **0.51±0.04** | 0.51 ± 0.04 | 0.51 ± 0.04 | 0.51 ± 0.04 | 0.51 ± 0.04 | 0.49 ± 0.04 |
| Sim8 | 0.45 ± 0.03 | 0.48 ± 0.03 | 0.49 ± 0.07 | 0.48 ± 0.07 | **0.50±0.07** | 0.47 ± 0.09 | 0.48 ± 0.08 | 0.49 ± 0.09 | 0.48 ± 0.09 | 0.48 ± 0.09 | 0.44 ± 0.07 |
| Sim10 | 0.48 ± 0.04 | 0.49 ± 0.03 | 0.51 ± 0.07 | 0.50 ± 0.08 | **0.51±0.08** | 0.48 ± 0.11 | 0.48 ± 0.10 | 0.49 ± 0.10 | 0.48 ± 0.11 | 0.48 ± 0.11 | 0.48 ± 0.09 |
| Sim11 | 0.49 ± 0.01 | 0.49 ± 0.01 | 0.50 ± 0.04 | 0.50 ± 0.05 | 0.50 ± 0.05 | 0.50 ± 0.05 | 0.50 ± 0.04 | 0.50 ± 0.05 | 0.50 ± 0.05 | 0.50 ± 0.05 | **0.51±0.08** |
| Sim12 | 0.48 ± 0.01 | 0.49 ± 0.01 | 0.50 ± 0.03 | 0.50 ± 0.05 | 0.49 ± 0.05 | 0.49 ± 0.05 | 0.50 ± 0.04 | 0.49 ± 0.05 | 0.49 ± 0.05 | 0.49 ± 0.05 | **0.52±0.06** |
| Sim13 | 0.49 ± 0.02 | 0.50 ± 0.02 | **0.52±0.06** | 0.51 ± 0.11 | 0.50 ± 0.10 | 0.50 ± 0.09 | 0.50 ± 0.09 | 0.49 ± 0.10 | 0.49 ± 0.10 | 0.49 ± 0.10 | 0.43 ± 0.10 |
| Sim14 | 0.48 ± 0.03 | **0.49±0.02** | 0.46 ± 0.06 | 0.46 ± 0.08 | 0.47 ± 0.07 | 0.44 ± 0.11 | 0.44 ± 0.10 | 0.45 ± 0.10 | 0.44 ± 0.10 | 0.44 ± 0.10 | 0.43 ± 0.07 |
| Sim15 | 0.44 ± 0.03 | 0.47 ± 0.04 | 0.49 ± 0.07 | 0.51 ± 0.10 | **0.51±0.09** | 0.49 ± 0.10 | 0.49 ± 0.09 | 0.50 ± 0.09 | 0.49 ± 0.09 | 0.49 ± 0.09 | 0.47 ± 0.07 |
| Sim16 | 0.51 ± 0.02 | 0.50 ± 0.02 | 0.50 ± 0.06 | 0.50 ± 0.07 | 0.50 ± 0.06 | 0.48 ± 0.07 | 0.49 ± 0.07 | 0.50 ± 0.08 | 0.49 ± 0.09 | 0.48 ± 0.09 | **0.53±0.09** |
| Sim17 | 0.48 ± 0.01 | 0.49 ± 0.01 | 0.49 ± 0.03 | 0.49 ± 0.05 | 0.49 ± 0.04 | 0.49 ± 0.04 | 0.49 ± 0.04 | 0.49 ± 0.04 | 0.49 ± 0.04 | 0.49 ± 0.05 | **0.56±0.06** |
| Sim18 | 0.49 ± 0.04 | 0.49 ± 0.03 | 0.49 ± 0.06 | 0.51 ± 0.08 | **0.51±0.08** | 0.48 ± 0.11 | 0.48 ± 0.11 | 0.48 ± 0.10 | 0.47 ± 0.11 | 0.47 ± 0.12 | 0.46 ± 0.09 |
| Sim21 | 0.48 ± 0.03 | 0.49 ± 0.02 | 0.50 ± 0.04 | 0.49 ± 0.07 | **0.51±0.08** | 0.47 ± 0.11 | 0.48 ± 0.10 | 0.48 ± 0.10 | 0.47 ± 0.11 | 0.47 ± 0.11 | 0.51 ± 0.07 |
| Sim22 | 0.48 ± 0.03 | 0.49 ± 0.02 | 0.50 ± 0.05 | 0.50 ± 0.06 | 0.51 ± 0.06 | 0.47 ± 0.11 | 0.46 ± 0.11 | 0.46 ± 0.11 | 0.45 ± 0.11 | 0.45 ± 0.11 | **0.51±0.09** |
| Sim23 | 0.40 ± 0.03 | 0.45 ± 0.05 | 0.48 ± 0.08 | 0.47 ± 0.09 | 0.48 ± 0.08 | 0.45 ± 0.11 | 0.46 ± 0.10 | 0.46 ± 0.10 | 0.46 ± 0.10 | 0.45 ± 0.11 | **0.51±0.06** |
| Sim24 | 0.40 ± 0.05 | 0.45 ± 0.06 | 0.49 ± 0.08 | 0.50 ± 0.10 | **0.50±0.09** | 0.47 ± 0.12 | 0.48 ± 0.11 | 0.49 ± 0.11 | 0.47 ± 0.12 | 0.47 ± 0.12 | 0.43 ± 0.08 |

Table 13: AUPRC Comparison on different simulations of NetSim under noise of scale 0.5 (mean ± std).

| Dataset | Corr | DYNO | GC | MI | PCMCI+ | LiNGAM | cMLP | cLSTM | eSRU | SRU | Transformer |
|---|---|---|---|---|---|---|---|---|---|---|---|
| Sim1 | $0.25 \pm 0.00$ | $0.25 \pm 0.00$ | $0.29 \pm 0.06$ | $0.28 \pm 0.06$ | $0.30 \pm 0.07$ | $0.29 \pm 0.07$ | $0.29 \pm 0.07$ | $0.29 \pm 0.07$ | $0.28 \pm 0.06$ | $0.28 \pm 0.06$ | **0.43±0.10** |
| Sim2 | $0.12 \pm 0.00$ | $0.12 \pm 0.00$ | $0.14 \pm 0.03$ | $0.14 \pm 0.03$ | $0.14 \pm 0.03$ | $0.14 \pm 0.03$ | $0.14 \pm 0.03$ | $0.14 \pm 0.03$ | $0.14 \pm 0.03$ | $0.14 \pm 0.03$ | **0.16±0.05** |
| Sim3 | $0.09 \pm 0.00$ | $0.09 \pm 0.00$ | $0.10 \pm 0.03$ | $0.10 \pm 0.03$ | $0.10 \pm 0.03$ | $0.11 \pm 0.03$ | $0.11 \pm 0.03$ | $0.10 \pm 0.03$ | $0.10 \pm 0.03$ | $0.10 \pm 0.03$ | **0.11±0.01** |
| Sim8 | $0.25 \pm 0.00$ | $0.25 \pm 0.00$ | $0.29 \pm 0.08$ | $0.29 \pm 0.07$ | $0.30 \pm 0.08$ | $0.29 \pm 0.07$ | $0.28 \pm 0.07$ | $0.29 \pm 0.07$ | $0.28 \pm 0.07$ | $0.28 \pm 0.07$ | **0.34±0.06** |
| Sim10 | $0.26 \pm 0.02$ | $0.25 \pm 0.01$ | $0.31 \pm 0.10$ | $0.30 \pm 0.09$ | $0.31 \pm 0.09$ | $0.30 \pm 0.09$ | $0.29 \pm 0.08$ | $0.29 \pm 0.08$ | $0.29 \pm 0.08$ | $0.29 \pm 0.07$ | **0.40±0.14** |
| Sim11 | $0.12 \pm 0.00$ | $0.12 \pm 0.00$ | $0.14 \pm 0.03$ | $0.14 \pm 0.04$ | $0.14 \pm 0.03$ | **0.14±0.04** | $0.14 \pm 0.03$ | $0.14 \pm 0.03$ | $0.14 \pm 0.03$ | $0.14 \pm 0.03$ | $0.14 \pm 0.03$ |
| Sim12 | $0.12 \pm 0.00$ | $0.12 \pm 0.00$ | $0.14 \pm 0.03$ | $0.14 \pm 0.03$ | $0.13 \pm 0.02$ | $0.14 \pm 0.03$ | $0.14 \pm 0.03$ | $0.13 \pm 0.03$ | $0.13 \pm 0.03$ | $0.13 \pm 0.03$ | **0.15±0.04** |
| Sim13 | $0.35 \pm 0.05$ | $0.36 \pm 0.05$ | $0.40 \pm 0.08$ | $0.41 \pm 0.09$ | $0.41 \pm 0.08$ | $0.40 \pm 0.08$ | $0.40 \pm 0.08$ | $0.39 \pm 0.08$ | $0.39 \pm 0.08$ | $0.39 \pm 0.08$ | **0.42±0.08** |
| Sim14 | $0.25 \pm 0.00$ | $0.25 \pm 0.00$ | $0.25 \pm 0.01$ | $0.26 \pm 0.03$ | $0.26 \pm 0.03$ | $0.25 \pm 0.04$ | $0.25 \pm 0.03$ | $0.25 \pm 0.03$ | $0.25 \pm 0.03$ | $0.25 \pm 0.04$ | **0.35±0.09** |
| Sim15 | $0.25 \pm 0.00$ | $0.25 \pm 0.00$ | $0.29 \pm 0.08$ | $0.30 \pm 0.08$ | $0.30 \pm 0.08$ | $0.29 \pm 0.08$ | $0.28 \pm 0.07$ | $0.28 \pm 0.07$ | $0.28 \pm 0.07$ | $0.28 \pm 0.07$ | **0.39±0.08** |
| Sim16 | $0.36 \pm 0.01$ | $0.36 \pm 0.01$ | $0.38 \pm 0.05$ | $0.38 \pm 0.05$ | $0.37 \pm 0.05$ | $0.37 \pm 0.05$ | $0.37 \pm 0.04$ | $0.37 \pm 0.05$ | $0.37 \pm 0.05$ | $0.37 \pm 0.05$ | **0.45±0.08** |
| Sim17 | $0.12 \pm 0.00$ | $0.12 \pm 0.00$ | $0.13 \pm 0.03$ | $0.14 \pm 0.03$ | $0.13 \pm 0.03$ | $0.14 \pm 0.03$ | $0.14 \pm 0.03$ | $0.13 \pm 0.03$ | $0.13 \pm 0.03$ | $0.13 \pm 0.03$ | **0.17±0.03** |
| Sim18 | $0.26 \pm 0.02$ | $0.26 \pm 0.02$ | $0.29 \pm 0.08$ | $0.31 \pm 0.08$ | $0.31 \pm 0.08$ | $0.29 \pm 0.08$ | $0.29 \pm 0.07$ | $0.28 \pm 0.07$ | $0.28 \pm 0.07$ | $0.28 \pm 0.07$ | **0.39±0.10** |
| Sim21 | $0.25 \pm 0.00$ | $0.25 \pm 0.00$ | $0.29 \pm 0.07$ | $0.29 \pm 0.07$ | $0.31 \pm 0.07$ | $0.29 \pm 0.07$ | $0.29 \pm 0.07$ | $0.28 \pm 0.07$ | $0.28 \pm 0.07$ | $0.28 \pm 0.07$ | **0.42±0.07** |
| Sim22 | $0.25 \pm 0.00$ | $0.25 \pm 0.00$ | $0.28 \pm 0.07$ | $0.29 \pm 0.07$ | $0.29 \pm 0.06$ | $0.27 \pm 0.06$ | $0.27 \pm 0.06$ | $0.27 \pm 0.06$ | $0.27 \pm 0.06$ | $0.26 \pm 0.06$ | **0.42±0.10** |
| Sim23 | $0.25 \pm 0.00$ | $0.25 \pm 0.00$ | $0.28 \pm 0.07$ | $0.28 \pm 0.07$ | $0.28 \pm 0.07$ | $0.27 \pm 0.07$ | $0.27 \pm 0.06$ | $0.27 \pm 0.06$ | $0.27 \pm 0.06$ | $0.26 \pm 0.06$ | **0.36±0.05** |
| Sim24 | $0.25 \pm 0.01$ | $0.25 \pm 0.01$ | $0.29 \pm 0.07$ | $0.30 \pm 0.08$ | $0.30 \pm 0.08$ | $0.29 \pm 0.08$ | $0.28 \pm 0.07$ | $0.29 \pm 0.07$ | $0.28 \pm 0.07$ | $0.28 \pm 0.07$ | **0.36±0.06** |

