# OpenReview forum: "Learning domain-specific causal discovery from time series"
_TMLR — Accepted by TMLR_

### Review · Reviewer_b4A9 · 2023-07-22

**Summary Of Contributions:**

The paper concerns causal discovery from time-series data, by using a transformer based classifier which had access to the 'ground truth' causality labels obtained using perturbation experiments. The approach is empirically evaluated on three problems, microprocessor simulation, FMRI simulated data and in silico gene networks temporal data. It showed improved predictive performance in terms of AUC and AUPRC, compared agains the approaches that does not utilize the ground truth labels.

**Audience:**

Yes

**Claims And Evidence:**

Yes

**Requested Changes:**

- Clarifying/correcting the claims that problem is not 'causal discovery from observational data', but from combination of observational time-series data and causation labels obtained from permutation tests
- Evaluating additional approaches that can also utilize the 'ground truth' labels
- Trying different noise type, or the same gaussian noise with higher strengths 0.3/0.5
- Proofread the manuscript text and fix typos


UPDATE
After revision updating the assessment

**Strengths And Weaknesses:**

Strengths:
-
- Inferring causality from time-series is very relevant research topic with potentially high impact
- The demonstrated empirical evaluation setup yield high predictive performance
- Explanation study using Gradient-weighted Class Activation Mapping is insightful

Weaknesses:
-
- Authors claim that the approach is solving the 'causal discovery from observational data', however the training labels for the classifier are obtained using the perturbation procedure, which is essentially an experiment with the treatment and control - and not observational data anymore.
- Even though the empirical evaluation shows high performance, especially agains the competitor baselines, the chosen baseline approaches are weak. The approaches like Correlation, Mutual Information, Granger Causality and Structural Equation Models are not utilizing the ground truth labels to which proposed approach had access. The approaches that relies on classifiers would be more appropriate competitors, especially neural network based ones. In addition, other transformer based causal inference approaches has been published before, for example:
1) Wang, Xingqiao, Xiaowei Xu, Weida Tong, Ruth Roberts, and Zhichao Liu. "InferBERT: a transformer-based causal inference framework for enhancing pharmacovigilance." Frontiers in Artificial Intelligence 4 (2021): 659622.
2) Melnychuk, Valentyn, Dennis Frauen, and Stefan Feuerriegel. "Causal transformer for estimating counterfactual outcomes." In International Conference on Machine Learning, pp. 15293-15329. PMLR, 2022.
3) Bi, Xiaotian, Deyang Wu, Daoxiong Xie, Huawei Ye, and Jinsong Zhao. "Large-scale chemical process causal discovery from big data with transformer-based deep learning." Process Safety and Environmental Protection 173 (2023): 163-177.
- The Evaluation under noise section, for the microprocessor example, could have been executed in a manner which would be more insightful. Transistor signals are binary, and adding the gaussian noise would make effect only if strong enough to flip the signal (0 to 1 or vice versa). Based on the table 2 results, it appears that even the strongest standard deviation of 0.1 was not flipping the signal often enough.
- The type of causality considered is quite specific/narrow. For example, limiting the cause effect consideration to only a short-time window, to avoid the challenging phenomena like feedback loops and long term/delayed effects.
- Some statements are vague, unclear or not supported:
1) Machine Learning approach is said to rely on 'minimal human assumptions' like iid. And I guess it is to contrast it to assumptions made in Granger Causality, that the cause precedes the effect. However, it seems rather subjective to attach the strength grade to assumption, or say that one is more/less strict than the other in general.
2) 'The high-resolution state recording is sparse and lengthy, which makes directly feeding it into the network less informative' - it is not clarified what measure of 'informative' is used, but complete signal definitely does not contain less information, as compared to truncated signal focused on 'periods where cause-effect relationships may be discernible'.
3) 'It is skeptical that the domain-agnostic causal discovery methods departing from human assumptions can still work well on it.' - this statement is unclear.

---

> ### Author Response · Authors · 2023-08-16
> **Reply to Reviewer b4A9**
>
> We appreciate your careful reading and thoughtful suggestions.
>
> > Authors claim that the approach is solving the 'causal discovery from observational data', however the training labels for the classifier are obtained using the perturbation procedure, which is essentially an experiment with the treatment and control - and not observational data anymore.
>
> A1: Thank you for pointing it out, the estimator is trained with observational and interventional data but only needs observational data during inference. We argue that it is doing causal discovery from observational data but clarified in the introduction of the manuscript that it does need perturbation data during training.
>
> > Even though the empirical evaluation shows high performance, especially agains the competitor baselines, the chosen baseline approaches are weak. The approaches like Correlation, Mutual Information, Granger Causality and Structural Equation Models are not utilizing the ground truth labels to which proposed approach had access. The approaches that relies on classifiers would be more appropriate competitors, especially neural network based ones. In addition, other transformer based causal inference approaches has been published before
>
> A2: First, while the comparison approaches we use are super simple, we want to point out that many if not most branches of science, including neuroscience and econometrics are dominated by them. We mentioned why we chose these baselines for MOS 6502 in Section 3.1, Appendix A1.2 and response to all reviewers. We would like to point out that more complex baselines (neural network based) are compared in the other experiments on small systems such as NetSim. Second, while relevant machine learning approaches exist, and in fact, the references 1-3 above are highly relevant, they work on very different problems to us, none of them uses causal ground truth to learn causal discovery. Using text to do causal inference is great, and we could also combine language models with our approach. It arguably is learned causal discovery because it is learned from text (which contains causal data) and then applied in context. The Causal Transformer is predicting counterfactual outcomes by leveraging Transformer to better process covariates, past treatments, and past outcomes, which is also a relevant but different topic. Bi, Xiaotian’s paper is highly relevant but also does not learn how to do causal discovery. It also now cites the Bi, Xiaotian’s paper for its contribution.
>
> > The Evaluation under noise section, for the microprocessor example, could have been executed in a manner which would be more insightful.
>
> A3: Thank you for providing additional experiment suggestions. In the noise experiment, there would be no flipping since the noised signal is not thresholded to a binary value. We added small-scale Gaussian noise here to simulate the small-scale observational noise. Gaussian noise here is added directly to the original binary value, which makes the noised signal a float value around 0 or 1. We have included the result with noise scales 0.3 and 0.5 for MOS 6502 and NetSim in the appendix.
>
> > The type of causality considered is quite specific/narrow. For example, limiting the cause effect consideration to only a short-time window, to avoid the challenging phenomena like feedback loops and long term/delayed effects.
>
> A4: As we mentioned in the manuscript, we are more interested in direct causal effect which is more likely to happen in a short-time window. We would attempt to tackle long-time and more indirect causal effects in future work.
>
> > Some statements are vague, unclear or not supported
>
> A5: Thank you for pointing out them, we have corrected and clarified all these statements.
>
> > Requested Changes
>
> A6: Thank you. We clarified our exact claim. For additional approaches that can utilize the ‘ground truth’ labels, to our knowledge, traditional causal discovery methods are relying on human assumptions which don’t require and don’t have ground truth. Also thank you for giving suggestions for additional experiments. For the noise strength, we have added Gaussian noise with higher strengths of 0.3 and 0.5 in the appendix and we also have proofread the manuscript and fixed typos.

---

### Review · Reviewer_uS8n · 2023-07-31

**Summary Of Contributions:**

The contribution is more or less unchanged from last notably with additional experimentation in particular on the NetSim data and also including a gene simulation dataset as well as improved comparisons for the latter two datasets to current state-of-the-art.

The authors consider causal discovery by learning as inferred using deep learning approaches considering three model specifications; long-short-term-memory units (LSTM), convolutional networks and transformers respectively to identify causal relations in time-series data. Their approach is contrasted simple correlation and mutual information based procedures as well as standard (linear) Granger causality and ICA-LinGAM finding the proposed approach substantially outperforming these methods on datasets of transistor interactions in the MOS 6502 processor when executing programmes on games. Notably, the procedure relies on supervised training to identify the structure of the causal graph by inferring causal structures from sequence encoded embeddings considering LSTM, temporal convolutional network (TCN) and Transformers. They further find their procedure to be robust to noise and variation in the causal graph when trained on one game and tested on others. They further examine their procedure on simulated fMRI based on the NetSim toolbox as well as the Dream3 silico gene network benchmark including more advanced causal discovery procedures including neural casual discovery methodologies. Model explainability is further considered by use of the Grad-CAM procedure.


**Audience:**

Yes

**Broader Impact Concerns:**

Whereas broader impact was not discussed in the previous submission I find the current discussion very superficial and very brief. Arguable the authors consider causal discovery in fairly harmless domains, however, discussions on limitations of such inferences, consequences of wrong inferences based on such modeling approaches as well as the importance of causal discovery as enabled by the proposed approach also for such discovery in larger systems would be good to further elaborate upon.


**Claims And Evidence:**

Yes

**Requested Changes:**

I find that the revised version has improved. The authors in the revised manuscript include in the introduction an overview of non-linear and neural causal discovery approaches. However, to make the paper self-contained it would be good to further detail how the procedures now compared to work more carefully.

Whereas the previous submission was in need of proof-reading I consider also this revised version in need of careful proof-reading. Below I provide some examples of where the text can be substantially improved:

“there are special aspects of time series.” Do you here mean:
“time-series provides additional means for causal discovery”

“We show test it on various domains” -> “We test it on various domains”

“on a part of microprocessor can be used to infer causal influences inside the other part of microprocessor.” -> “on a part of the microprocessor can be used to infer causal influences inside the other part of the microprocessor.”

“the minimal components consisting the causal graph” something is missing in this sentence structure. Do you mean “for the minimal component of the causal graph” – and what is meant here by minimal component?

“It is skeptical that the domain-agnostic causal discovery methods departing from human assumptions can still work well on it.” What is here meant by skeptical? – do you mean critical?

The following sentence is unclear to me:
“Interestingly, we find AUROC is not always suitable for causal discovery evaluation, since the causal components in real world are usually rare. Especially when we are dealing with extremely imbalanced dataset such as MOS 6502, AUROC will be biased toward the majority negative class and ignore the minority positive class”
A merit of the AUROC is that it is invariant to class imbalance as the true-postive and false positive rates are normalized respectively by the amount of positive and negative samples. Please clarify what you here mean. I do agree that AUCPR focuses  on the positive (rare) class as also discussed.

“Learned estimator can generalized on different behaviors” this should be “Learned estimator can be generalized to different behaviors”

“It shows that even our estimator” -> “It shows that despite that our estimator”

“on whether AUROC or AUPRC,” -> “both for AUROC and AUPRC,”

“For instance, the wire used to sample the voltages of transistors might be disrupted by the wind while recording brain waves.” What do you here mean by disrupted by the wind?

When stating:
“This benefits from the introduced supervised signal, which, arguably, makes our model capable of uncovering causality under noise while methods based on human intuition failed to extract effective features.” I am not sure this specifically should be attributed to the supervised signal but more so to the transformer architecture effectively denoising the signal. This should be further elaborated.

“to compare the multi-variate and graph-based discovery methods with us.” -> “to compare the multi-variate and graph-based discovery methods with our proposed approach.”

“Each network is consisted of 100 nodes” -> “Each network consisted of 100 nodes”

“The success of learning domian-specific” -> “The success of learning domain-specific”


**Strengths And Weaknesses:**

I have below updated my evaluations of the strengths and weaknesses in this revised version - and find that the authors in general have reasonably addressed my previous concerns in particular by the inclusion of the Dream3 dataset as well as neural and non-linear comparison procedures. I therefore find that the revised version more sufficiently include comparisons to current-state-of-the-art. The authors promise to provide code but this is not currently assessible due to anonymization, I have thus removed this request as well as minor requests.

Strengths:
• The proposed use of deep learning for causal discovery is valid and interesting.
• The defined causal graphs based on transistor networks form a to my understanding novel and interesting testing ground for evaluating methods for causal discovery based on time-series data of general utility.
• The paper is evaluated considering generalization across games as well as in terms of robustness to noise well probing the developed procedures with improved comparison to state of the art for the NetSim data and new inclusion of the Dream3 gene data.

Weakness:
• Why are the non-linear causality procedures included not explored also for the simulated MOS 6502 system?
• There are many benchmarks data for causal discovery and it is unclear why the developed procedure is not also evaluated on these benchmarks (further clarifying why this has not been considered would be good)

I think the authors have adjusted to some extent claims and evidences as previously requested as well as the title to somewhat better reflect the procedure. In particular, I find substantial improvements to the experimentation including further comparisons to state-of-the-art as previously requested. I have therefore stated yes below to claim and evidences - but the approach relying on supervised learning of causal pairs I still consider with strong limitations.

---

> ### Author Response · Authors · 2023-08-16
> **Reply to Reviewer uS8n**
>
> We thank you for your valuable suggestions and feedback in reviewing paper. We have uploaded code in the supplemental materials. It is worth noting that we also added evaluation across different time (periods) for MOS 6502 and in the resubmitted version.
>
> > Why are the non-linear causality procedures included not explored also for the simulated MOS 6502 system? • There are many benchmarks data for causal discovery and it is unclear why the developed procedure is not also evaluated on these benchmarks (further clarifying why this has not been considered would be good)
>
> A1: Thank you for proving experiments suggestions on other benchmarks. NetSim and Dream3 datasets are often used to benchmark causal discovery for time-series data, Considering most works didn’t consider AUPRC as an evaluation criterion, we used their settings and reran on these two datasets instead of directly using their results.
>
> > I find that the revised version has improved. The authors in the revised manuscript include in the introduction an overview of non-linear and neural causal discovery approaches. However, to make the paper self-contained it would be good to further detail how the procedures now compared to work more carefully.
>
> A2: We now clarify how we compare all methods with regard to their ability to predict test set causal ground truth effects based on observational data (and like all supervised studies from ground truth on the training set)
>
> > Below I provide some examples of where the text can be substantially improved
>
> A3: You went above and beyond at helping us sharpen the language. We are truly thankful and want to just let you know how much we appreciate this. We fixed all these mistakes.
>
> > Whereas broader impact was not discussed in the previous submission I find the current discussion very superficial and very brief. Arguable the authors consider causal discovery in fairly harmless domains...
>
> A4: We added an extended discussion of the broader impact issues. We now also discuss how malicious actors can benefit from causal discovery.

---

### Review · Reviewer_GUY6 · 2023-08-01

**Summary Of Contributions:**

The paper investigates the task of domain-specific causal discovery for time-series data. In contrast to other approaches that rely on human expertise, the paper studies the scenario where the causal discovery is data-driven and supervised. The paper presents empirical results on multiple datasets to demonstrate strong performance of the model over known baselines.

**Audience:**

Yes

**Broader Impact Concerns:**

No broad impact concerns

**Claims And Evidence:**

Yes

**Requested Changes:**

See above for specific notes about the clarity of the notation, figures, and tables.

**Strengths And Weaknesses:**

Strengths:

S1) I appreciate the empirical results, and the results from Table 1-5 suggest that the proposed model can improve on existing baselines.

S2) The breadth of the experiments is impressive, and the differences in noise settings, dataset sources, and dataset size is impressive.

Weaknesses:

W1) The paper's notation was often not defined. For example, the text before Eq1 never defines the do-notation or introduces p.  Also, Algorithm 1 is unclear. Where does the k subscript comes from in period k (line 4)? In Algorithm 2, is P a random variable, prediction, or something else?

W2) I found the figures and tables of the paper difficult to read. More explanation or better labeling would greatly strengthen the paper
 - Fig 3 and Fig 4 need x-axis labels for all three figures. The description in 3.1 was hard for me to follow along. Better figure-making or an annotated image would be much easier to read.
 - Table 1: would it be possible to get a confidence interval?
 - Table 1 needs explanation of which models are what. Section 3.2 would also be a good place to put the model abbreviations.

Additional notes
 - Typo page 10: "(see Table 1. Interestingly," -> needs closing parentheses.

---

> ### Author Response · Authors · 2023-08-16
> **Reply to Reviewer GUY6**
>
> We would like to thank you for carefully reviewing the paper and providing thoughtful advice.
>
> >  The paper's notation was often not defined. For example, the text before Eq1 never defines the do-notation or introduces p. Also, Algorithm 1 is unclear. Where does the k subscript comes from in period k (line 4)? In Algorithm 2, is P a random variable, prediction, or something else?
>
> A1: Thank you for pointing out the notation issues, we have added definitions about Eq1 and fix typos about $m$ in Algorithm 1. As we noted in Algorithm 2, P is the probability of causal relationship given observational data $X_n$, estimated by the estimator whose parameter is $\theta$.
>
> > I found the figures and tables of the paper difficult to read. More explanation or better labeling would greatly strengthen the paper
>
> A2: Thank you for the suggestions on the figures and tables. The x-axis in Fig 3 and Fig 4 represents the horizontal position of the transistor on the microprocessor, whose labels are supposed to be the ‘X position’ as we labeled. We added more clarifications in the captions of these two figures to help to understand. We also added explanations in Section 3.1 to help follow along. We added explanations for Table 1’s models in Section 3.2 as required. For Table 1, we added mean and standard deviation across 5 periods, which provides information about uncertainty. We also completed further proofreading to improve clarity.

---

### Author Response · Authors · 2023-08-16
**Manuscript Revision**

We thank for all reviewers’ valuable feedback and comments. We have answered each reviewer directly as replies and we also have made revisions to our manuscript accordingly.

We have made changes we would like to hightlight as follows:
* We clarified/corrected statements and claims that reviewers thought is not clear or vague.
* We proofread and fixed typos in the manuscript, added explanations, and adjusted tables and figures following reviewers’ suggestions.
* We added additional experiments and analysis about evaluation on MOS 6502 and NetSim under a noise scale of 0.3 and 0.5 in the Appendix Section.
* We added an extended discussion in broader impact concerns discussing challenges our approach will face.

We would like to point out that as we mentioned in the discussion, a reason we prototype our method on the microprocessor is simply due to a lack of data in other domains: neither medicine nor neuroscience has large datasets of observational data along with ground truth perturbation-based causality data. And we would like to explore non-linear methods for it but MOS 6502 is a very big system including hundreds of nodes, which is very time-consuming for neural network based (most non-linear causality procedures) and graph-based methods. Therefore, temporarily, we are only able to compare it with some fast pair-wise methods on MOS 6502, which are often to be linear.

We appreciate all the advice and are open to any follow-up suggestions about the manuscript.

---

### Decision · Action_Editors · 2023-09-10

**Recommendation:** Accept as is

**Comment:**

The paper is good as-is and has been thoroughly discussed by the reviewers.

**Audience:**

This is a paper for a wide audience.

**Claims And Evidence:**

The claims are indeed supported by accurate, convincing and clear evidence.
More on detail, experimental evidence remains limited by the low number of benchmarks are available ion the topic of causality learning. There is obviously a tension between real-word examples on one hand, for which the ground truth is hard to assess and toy datasets, on the other hand, that carry no ambiguity but have little practical relevance. The authors chose to consider few, but different and real-world examples. This choice makes sense to the reviewers.